# Assessment of extreme flood events in changing climate for a long-term planning of socio-economic infrastructure in the Russian Arctic

**E. Shevnina[1, 2], E. Kourzeneva[1], V. Kovalenko[2], and T. Vihma[1].**

[1]Finnish Meteorological Institute, P.O. Box 503 FI-0010, Helsinki, Finland

[2]Russian State Hydrometeorological University, Malookhtinsky prospect 98, 195196, Sankt-Petersburg, Russia

*Correspondence to*: E. Shevnina (elena.shevnina@fmi.fi)

**Abstract.** Climate warming has been and is expected to continue to be more acute in the Arctic than at lower latitudes, which generates major challenges for economic activity in the region. Among others issues is the long-term planning and development of socio-economic infrastructure (dams, bridges, roads, etc.), which requires climate-based forecasts of the frequency and magnitude of detrimental flood events. To estimate the cost of the infrastructure and operational risk in affected regime, a probabilistic form of long-term forecasting is preferable. In this study, a probabilistic model allowing simulation of a parameters of the probability density function (PDF) of multi-year runoff based on a projected climatology is applied to evaluate changes in extreme values of the flood depth of runoff for the territory of the Russian Arctic. The model is validated by cross-comparison of modeled and empirical PDFs using the observations on 23 sites located in the northern Russia. The mean values and coefficients of variation ($C_v$) of the flood depth of runoff are evaluated under four climate scenarios using the simulation results from six climate models for the period of 2010–2039. Regions with the expected substantial changes in means and $C_v$ of the flood depth of runoff are outlined. It is suggested to account for the future climate projections in the hydrological calculations of maximal discharges of rare occurancy for the sites located within such regions. The example of engineering calculations of the maximal discharge with 1 % exceedence probability is provided for the Nadym River at Nadym City.

## 1. Introduction

The economic importance of the Arctic is an increasingly recognized issue. Various governmental and commercial projects have been initiated internationally to develop the socio-economic infrastructure in the Arctic. Among others, there are projects for the important oil and gas fields in Mackenzie Valley, Canada (www.mackenziegasproject.com), Prudhoe Bay, USA (http://petrowiki.org/Prudhoe_Bay_field), as well as the Pechora and Yamal regions, Russia (www.gazprom.com/about/production/projects/mega-yamal/). To design hydraulic constructions, such as dams, bridges, roads and pipelines, and to estimate the costs and risks of flood damaging during the infrastructure's lifetime, an information is needed on threshold values of dangerous river discharges. These values are calculated from the upper-tail of PDFs of the maximal river runoff. These PDFs are usually modeled with three parametric distributions (e.g. Pearson type III or Log Pearson type III) using mean value,

the coefficient of variation and coefficient of skewness (SP33-101-2003, 2004; Bulletin 17–B, 1988). These parameters are calculated from observations with an assumption of stationarity of climate and hydrological regime (Thomas, 1985). It means that the values of the PDFs parameters do not change in the future or during the period of operation of building constructions.

A great number of weather anomalies and detrimental flooding events are observed during the last decade. The changes in climate are especially expressed in the polar regions. Climate models project a robust increase in precipitation over the Arctic and sub-Arctic (Collins et al., 2013; Laine et al., 2014). During October through March, precipitation in the Arctic is expected to increase by 35 % and 60 %, under medium and high greenhouse gas concentration pathways, respectively (RCP4.5 and 8.5), relative to the period 1986–2005 (IPCC, 2013). The

projected precipitation increases in April through September are 15 % and 30 %, respectively. Due to climate warming and increased rainfall, annual-mean snowfall is projected to decrease over northern Europe and mid-latitude Asia, but to increase in northern Siberia, especially in winter (Krasting et al., 2013). Further, precipitation extremes are projected to increase, the climate model results being robust particularly for northern Eurasia in winter (Kharin et al., 2013; Toreti et al., 2013; Sillman et al., 2013). In Siberia these increases in

precipitation will be accompanied by a decrease in the number of consecutive dry days (Sillman et al., 2013). Over northern Eurasia, the net precipitation (precipitation minus evapotranspiration) is also projected to increase during winter. The projected changes discussed above are likely with a high confidence (Collins et al., 2013), and therefore point to an urgent requirement to better evaluate the response of the other components of the Arctic freshwater system, including terrestrial hydrology (Prowse et al., 2015).

There are two opposite opinions about the climate changes and their effects to hydrological regime to answer the question: "Is it necessary to account for climate changes by water managers and stakeholders?". According to Milly et al. (2008) the climate effects are already substantial, and should be taken into account by planers and water managers. The opposing view doubts a climate-driven changes and calls to attention the uncertainties due to short observed time series (Lins and Cohn, 2011; Montanari and Koutsoyiannis 2014; Serinaldi and Kilsby,

2015). We propose to account for the future climate changes and their impacts on environmental risks even in case if the uncertainties are unknown. In the other words, it is better to prevent an accident than to deal with its consequences, which may be more expensive than the initial investment. We consider that the changes in meteorological variables would remain noticed in runoff, which is an element of general water balance. From a practical point of view, a method to evaluate of the extreme flood events based on climate scenarios is required

irrespective of debates about the extent of reality of the changes.

Two approaches are usually applied in the evaluation of the future maximal runoff values of rare occurrence. The physically-based approach is based on a combined use of regional climate model (RCM) and rainfall-runoff hydrological model (Fig. 1). RCMs provide the future meteorological forcing variables with a high temporal resolution to drive a hydrological model that describes complex physical processes, such as infiltration, snow

melting, and evapotranspiration. This allows generation of synthetic time series of river runoff (discharges) for individual watersheds (Archeimer and Lindström, 2015). The flood events with required exceedance probability are then estimated from the simulated time series. Successful applications of this approach include numerous studies (Veijalainen et al., 2010; Lawrence et al., 2011). The large-scale rainfall-runoff models have also been used to assess the changes in the flood frequency by Lehner et al. (2006) for the European Arctic. One

shortcoming of these studies is that the resulting flood frequency estimations are sensitive to the algorithms of the calculation of a pseudo-daily precipitation input from projected climatology provided by Global Circulation Models (GCMs) (Verzano 2009). The second approach to evaluating the hydrological response to the expected climate change is stochastic (Fig. 1). The stochastic components are incorporated into the physically-based hydrological model (Kuchment and Gelfan, 2011) to generate the runoff time series based on the statistics of meteorological variables (weather generators). Thus, estimates of the extreme hydrological events (floods or droughts) with the required exceedance probability could be obtained for any climate scenario by producing the meteorological signal with Monte-Carlo method. Both approaches are usually applied for a single catchment. In performing regional scale flood (or drought) analysis, the runoff signal should be simulated for a set of watersheds. It makes the calculations extremely costly computationally, especially in the case of climate ensembles.

The approach presented in this paper could be named as probabilistic (to distinguish from the stochastic modelling described above). This approach allows us to skip the generation of the runoff time series, since only PDF parameters are directly calculated from the meteorological statistics for the projected periods of 20–30 years (Fig. 1). These simulated PDF parameters are further used to evaluate the future runoff values with the required exceedance probability using theoretical distributions i.e. from the Pearson system (Elderton et al., 1969). Since the probabilistic model simulates only three-four parameters of PDF, this approach allows to perform the regional scale assessment of the detrimental hydrological events in the future, and to difine the regions where the risks of damage to infrastructure are increase.

In the presented study, we consider that for the Arctic, the maximal runoff is formed during a spring flooding. The Pearson type III distribution is used to model the PDF of the spring flood depth of runoff and to estimate the maximal discharge with the required exceedance probability. The probabilistic approach used in this study combines the statistical methods and elements of theory of Markov processes. Both of them are traditionally applied in hydrological engineering calculations to evaluate hydrological extremes (Kite, 1977; Benson, 1968; Kritsky and Menkel, 1946). The traditional analysis of flood and drought frequency requires the hydrological time series to estimate the parameters of the PDFs. However, the parameters of PDFs can be also estimated from the statistics of meteorological variables. The idea to perform the direct simulation of the PDFs' parameters from climate projections (without the simulation of time series) is proposed by Kovalenko (1993). Kovalenko et al. (2010) simplified the basic probabilistic model for engineering hydrology, and Viktorova and Gromova (2010) applied this approach to produce a regional-scale assessment of the future drought extremes for the European part of Russia.

The main idea of the simplified method is the "quasi-stationarity" of the changing climate and hydrological regime for the periods of 20–30 years. This idea allows us to represent the multi-year runoff statistically with a set of PDFs' parameters for the particular time window; the set is different for the past (or reference period) and the future (or projected period) climatology. Thus, the climate changes could be accounted in calculations of the runoff tailed values, which are usually required for risks assessment in water management. The IPCC recommends the climate projections, which are represented as the multi-year means of the meteorological values for the period of 20–30 years (Pachauri and Reisinger, 2007), i.e. under the same quasi-stationarity assumption. This probabilistic model provides a more economical way to produce hydrological projections for the extremes on a regional scale. This is because of (i) a low number of forcing and simulated variables (only three-four statistics of

climate and hydrological variables are needed); (ii) a low number of parameters (physical processes described integrally by a lumped hydrological model); and (iii) a relative simplicity of a regionally-oriented parameterization. Furthermore, the probabilistic model does not require large spatially distributed datasets and may be applied for regions of poor data coverage, such as the Arctic.

The aim of this study is to perform a regional-scale assessment of the future extreme flood events based on climate projections for the Russian Arctic. The novelty of the study includes two aspects. First, we present a method to assess the frequency and magnitude of extreme flood events in changing climate, adapted in this case to the Arctic territories. It could also be applied to other territories , as the regionally oriented parameterization is relatively simple. Second, the paper provides the projected changes in the mean values and $C_v$ of the flood depth of runoff under the four climate

scenarios for the Russian Arctic. The regional-scale assessment of the statistics of the flood depths of runoff is based on the Special Report on Emissions Scenarios (SRES) and Representative Concentration Pathway (RCP) scenarios. The regions are delineated where the frequency and magnitude of the floods are expected to change substantially. These maps include a warning for the regions where the engineering calculations of the extreme maximal discharges should be corrected to account for the climate changes. An example of the engineering calculation of maximal

discharge of 1% exceedance probability for the Nadym River at Nadym City is provided using the outputs of three climate models for the period 2010–2039.

## 2. Methods and data

The idea of the method is (i) to simulate the future parameters of PDF of the multi-year runoff using the projected mean values of precipitation and air temperature. Then, (ii) to construct the PDF with simulated parameters and a

priory defined theoretical distribution (Pearson Type III), and finally (iii) to calculate the maximal runoff from tailed values with required probability of exceedance. This idea is used to perform the regional-scale assessment of the maximal extremes for the northern territories, where these values occur during the spring floods. Within these territories a peak flow is usually formed by seasonal snow melting and represented by a spring flood depth of runoff ($h$, mm/(time period)), calculated as the volume of spring flood flow ($m^3$) from the drainage basin divided by its area

($m^2$). The reason, why the value of spring flood depth of runoff was chosen instead of the maximal discharge, is that this value allows mapping of the spatial distribution of a river's maximal flow over broad areas. Thus, the value of spring flood depth of runoff can be used in defining the regions for which the flood extreme maximum discharge should be corrected according to climate change. After such regions were delineated, the correction of maximal discharge with required probability of exceedance can be done using climate projections based on the historical

discharge time series (for the watersheds with observations) as well as based on the mapped projected mean value, $C_v$ and $C_s$ of the spring flood depth of runoff (for the catchments without observations). In this case, the extreme river discharge ($Q$, $m^3s^{-1}$) with a required probability of exceedance ($p$) is calculated according method proposed in SP33-101-2003 (2004):

$$Q_p = k_0 \mu h_p \delta \delta_1 \delta_2 F / (F+b)^n \ , (1)$$

where $k_0$ is flood coincidence factor, which reflects a simultaneousness of precipitation/melting water input, i.e. depends on the shape of the hydrograph; $\mu$ is a factor of inequality of the depth of runoff and maximal discharge

statistics; $h_p$ is a spring flood depth of runoff (mm/(time period)) with probability $p$ (0.1, 0.05, 0.01) estimated from an exceedance probability curve (or PDF); $\delta, \delta_1, \delta_2$ are watershed fractions of lake, forest and swamp respectively; $F$ is a watershed area (km²); $b$ is the additional area which adjusts the reduction of the runoff (km²) and $n$ is degree of a runoff reduction. For the ungauged basin the value of $k_0$ is estimated from observations on a neighboring gauge located on a same type of landscape (SP33-101-2003, 2004). In our study, the value of $k_0$ was considered to be constant for the reference and projected periods. The values of $\mu, \delta, \delta_1, \delta_2$, $b$ and $n$ may be obtained from look-up tables (SP33-101-2003, 2004) or from global datasets representing land cover (e.g. Bertholomee´ and Belward, 2005). To estimate the spring flood depth of runoff with required probability of exceedance ($h_p$), the PDF is constructed based on the mean value, $C_v$ and $C_s$. These values are calculated from the observed time series, but in our study we simulate them based on the projected climatology for the future time period 2010–2039.

## 2.1 Model

The core of the probabilistic hydrological model is a linear differential equation with stochastic components having solutions statistically equivalent to solutions of the Fokker–Planck–Kolmogorov (FPK) equation. It allows the evaluation of the probability density function of a random hydrological variable with parameters dependent on climate variables (Kovalenko, 2014, 1993). Under a quasi-stationary assumption of the climate change, the FPK is approximated by a system of algebraic equations to simulate initial statistical moments of multi-year runoff. These moments are further used to calculate the PDFs' parameters and to model them using the theoretical formulations (e.g. Pearson Type III). In our study, the simple model suggested in (Kovalenko et al., 2010, see the Annex for details) was used to model the first and second statistical moments of the flood depth of runoff:

$$
\begin{aligned}
-\bar{c}\, m_1 + \bar{N} &= 0 \\
-2\,\bar{c}\, m_2 + 2\,\bar{N}\, m_1 + G_{\widetilde{N}} &= 0
\end{aligned}
\quad , (2)
$$

where $m_1$ (mm) and $m_2$ (mm²) are the first and second statistical moments of the flood depth of runoff for the period of 20–30 years; $\bar{c} = 1/k\tau$ is inverse of the runoff coefficient $k$ (which is a dimensionless coefficient, the ratio of the amount of runoff to the amount of precipitation received) times the watershed reaction delay $(\tau)$; $\bar{N}$ (mm) is the mean value of the annual precipitation amount for a period of 20–30 years. The parameter $G_{\widetilde{N}}$ (mm²) is the variance of the annual precipitation amount.

The model (2) allows evaluating the multi-year runoff statistical moments for the projected time period based on the climatology and multi-year runoff statistics for the reference (historical) period. The climate and runoff regime are steady within both the reference and projected periods (the assumption of quasi-stationarity). The "steady" is defined statistically, i.e. there are no significant trends and changes in mean values of meteorological and hydrological characteristics within the periods. However, the basic statistics (mean, $C_v$ and coefficients of skewness $C_s$) are significantly different for the reference and projected periods.

The system of Eq. 2 was applied as follows:

− (i) to estimate the statistical moments from the observed hydrological and meteorological time series for the chosen reference (r) period ($m_{1r}$, $m_{2r}$ and $\bar{N}_r$ );

− (ii) to assess the model parameters for the reference period:

$$\bar{c}_r = \bar{N}_r / m_{1r} \ ,$$

$$G_{\widetilde{N}r} = 2\left(\bar{c}_r m_{2r} - \bar{N}_r m_{1r}\right) \ , (3);$$

− (iii) to calculate the future (f) values of two statistical moments ($m_{1f}$ and $m_{2f}$) from the projected mean of the annual precipitation $\left(\bar{N}_f\right)$, provided that the future parameter values ($\bar{c}_f$ and $G_{\widetilde{N}f}$ ) are known:

$$m_{1f} = \bar{N}_f / \bar{c}_f \ ,$$

$$m_{2f} = \left(2\bar{N}_f m_{1f} + G_{\widetilde{N}f}\right)/2\bar{c}_f \ , (4).$$

The values of the parameters $\bar{c}$ and $G_{\widetilde{N}}$ either can be set constant for the projected time period as proposed by Kovalenko et al. (2010) or depending on the future climatology. To evaluate the projected values of the parameter $\bar{c}$ depending on the average precipitation and air temperature, the linear equations are suggested by Shevnina (2012). In this study, both methods of handling these parameters are considered.

− (iv) to obtain the future statistical values of the spring flood depth of runoff: the mean value and $C_v$. The future $C_s$ was calculated from the given ratio of $C_s/C_v$ which is considered to be constant for the reference and future periods. The future PDFs were constructed with Pearson type III theoretical distributions based on these statistical values and used to estimate the spring flood flow depth of runoff with the required exceedance probability. Then, the maximal flood discharges were calculated using Eq. 1.

**2.2 Validation**

Rainfall-runoff models are usually validated against observed time series (Lehner et al., 2006; Arheimer and Lindström, 2015). The system of Eq. 2 allows simulating the parameters of PDF of the multi-year runoff without producing time series. These predicted parameters of the PDFs for the one time period are based on the parameters of PDFs calculated for the other period. Two time periods should have the different PDFs' parameter values and this difference should be statistically significant (Kovalenko et al., 2010). Such kind of periods were found in the observed time series to perform the probabilistic model validation using a cross-validation procedure. In the simplest cross-validation procedure, the dataset of measurements (observations) is separated into two sub-sets, called the training set and the testing/control set. The training set is used to evaluate the model parameters, which are further used to calculate the nominally predicted values of the parameters of the control PDFs. In our case, the nominally predicted PDF was compared with the empirical distribution for the testing/control set using statistical goodness-of-fit tests.

For the period of observations, sub-periods with a statistically significant difference (shift) in the mean values were selected. The shifts in the subsampled mean values (corresponding to the sub-periods) were detected

according to the Student's *t*-test using the moving window approach (Ducré-Robitaille et al., 2003). We begin from setting the size of the first subsample to the chosen minimum (15 members) and calculating the value of *t*-test. The size of the second subsample is taken as the size of the total sample (*N*) minus the chosen minimum ([*N*-15] in Fig. 2) in this case. Then, the size of the first subsample was incremented by an iterator *i*=1, 2, 3 … until the size of the second subsample is equal to the chosen minimum. The values of *t*-test were calculated for each step and were linked to the years of the time series subdivision. Finally, the time series was divided by the year having the value of *t*-test exceeding the critical value 0.05 level of statistical significance. The Student's test critical values accounting the asymmetry and autocorrelation in hydrological time series were used (Rogdestvenskiy and Saharyuk, 1981). If several partitioning years were recognized, we gave preference to the year that divided the time series into two approximately equal sub-periods.

The first and second statistical moments of the flood depth of runoff for each sub-periods were calculated according to Bowman and Shenton (1998). The third moment was estimated from the entire time series and the constant ratio of $C_s/C_v$ was calculated. The mean values of the annual precipitation and air temperature for each sub-period were also calculated (Table 1). The resulting dataset included pairs of the statistical moments for the flood depth of runoff ($m_1^I$, $m_1^{II}$, $m_2^I$, $m_2^{II}$), the mean values of air temperature $\left( \bar{T}^I, \bar{T}^{II} \right)$ and annual precipitation $\left( \bar{N}^I, \bar{N}^{II} \right)$.

For the cross-validation, we: (i) considered the first sub-period as the training and calculated the reference values of the model parameters; (ii) predicted nominally ("in the past") the first and second moments for the second sub-period (which was considered as control). The same procedure was applied backwards. For the period of the nominal prediction two model versions were considered: (i) with the basic parameters setting as proposed by Kovalenko et al. (2010) and (ii) with the regional-oriented parameterization as suggested by Shevnina (2012). In our study, the parameter $G_{\widetilde{N}}$ was considered to be constant for the projected time period. The mean values and the coefficients of variation were calculated with the nominally predicted statistical moments and the coefficients of skewness were estimated from the constant ratio of $C_s/C_v$ for each time sub-period. Then, the multi-year PDFs of flood depth of runoff were modeled with Pearson type III distribution using the nominally predicted mean, $C_v$ and $C_s$. The empirical probability distribution and nominally predicted PDF were compared for each sub-period and the goodness-of-fit between them was estimated using Pearson chi-squared and Kolmogorov-Smirnov one-sample tests. If the value of the test did not exceed the critical value of 0.05 level of statistical significance, it was considered to be successful in regard to the nominal prediction of the statistical moments. The model's prediction scores were estimated as a percentage of matching PDFs estimated from the whole dataset (Table 1).

An example of the cross-validation is given for the Yana River at the Verkhoyansk gauge (Fig. 2). In order to partition the flood depth of runoff time series into two sub-periods, the time series (Fig. 2, top) was first divided at the point S=1949 and the first *t*-test value was calculated. Then the *t*-test values were calculated step-by-step until the point E=1987 with increments of 1 year. At the point A=1965 (Fig. 2, bottom), the *t*-test value exceeds the *t*-critical value at 0.05 level of statistical significance. Thus, two periods were differentiated: the first sub-period, covering the interval 1935–1964 with $m_1^I$=41.2 (mm) and the second sub-period covering the interval 1965–2002 with $m_1^{II}$=52.3 (mm). The second statistical moments ($m_2^I$, $m_2^{II}$) of each period were also calculated.

Then, the mean values of the annual precipitation amount $\left( \bar{N}^I, \bar{N}^{II} \right)$ and the annual average air temperature $\left( \bar{T}^I, \bar{T}^{II} \right)$ were also calculated for the two sub-periods. The reference values of the parameters $\left( \bar{c}_r, G_{\tilde{N}r} \right)$ were estimated using $m_1{}^I{}_r$, $m_2{}^I{}_r$ and $\bar{N}_r^I$ for the sub-period 1935–1964 (considered as training). Then, the nominally predicted or modeled $m_1{}^{II}{}_f$, $m_2{}^{II}{}_f$ were calculated from $\bar{N}_f^{II}$ for the sub-period 1965–2002

(considered as control). Finally, the nominally predicted mean value and ==$C_v$ of the== flood depth of runoff were calculated from the ==simulated runoff statistics== and ==$C_s$== was estimated from the ratio of $C_s/C_v$ for each period. These values were used to model the nominally predicted PDFs (or exceedance probability curves – Fig. 3) with the Pearson type III distribution. Then, the nominally predicted PDFs and empirical distribution were compared (Fig. 3). The same procedure was done backwards: the sub-period 1965–2002 was considered as training and the

statistical moments were nominally predicted for the sub-period 1935–1964 (considered as control in this case).
The model cross-validation was performed with observations collected during the period from 1930s to 2000s. The observed data were extracted from the official edition of the Multi-Years/Year Books of the State Water Cadastre of the Russian Federation (see e.g. Kuznetsov, 1966). The spring flood depth of runoff time series at 76 gauges for medium size catchments (1,000–50,000 km$^2$) were used. The gauges are located on the territory of the

Russian Arctic. The gauging sites are irregularly distributed over the territory with 65 % of the points located at the west part of the Arctic. The time series lengths vary from 26 to 77 years with an average of 51 years. The dataset has no gaps at the time series of 66 % of the considered gauges and for the time series of 18 % of the gauges have the missing values for more than 5 % of their length.
The sub-periods with statistically significant shift in the mean values of the flood depth of runoff were selected

for 23 time series (Table 1), which is 30 % of the considered data. For the corresponding watersheds, the mean values of the annual precipitation amount and the average air temperature were calculated using the observations for 37 meteorological stations (approximately 2 stations per watershed) for each sub-period (Table 1). The observed time series of the annual precipitation amount and the average air temperature for the meteorological sites were obtained from Razuvaev et al. (1993), Radionov and Fetterer (2003), Bryazgin N. (2008, personal

communication) and the multi-year catalogs of climatology (e.g. 1989).
For each gauge and sub-period the statistical moments were nominally predicted using Eq. 4 for two methods of the model's parameters settings (Table 2). Also, the statistical moments were considered to be constant during the entire observed period. In this case, the nominally predicted PDF for one sub-period was modeled using the statistical values calculated from observed data of the other sub-period ("no model" case in Table 3). The "no model" case

illustrates the scenario in which climate change is not taken into account, and thus the PDFs' parameters are not modified for the period of prediction. This case reflects the situation as considered in the guidelines for the engineering hydrology (SP33-101-2003, 2004; Bulletin 17–B, 1982), which used only observed time series to evaluate the PDF parameters. The percentage of the PDFs that matched successful to empirical PDFs according to Pearson chi-squared and Kolmogorov-Smirnov one-sample tests were evaluated for each version of the nominal prediction. Table 3

provides the percentage of the successful coincidences of the PDFs, which was obtained for whole available cross-validation dataset (46 pairs of the simulation and empirical PDFs).

The model using the constant parameters gives a more conforming result than the case of no model: the percentage of successfully matched PDFs is over 5–10 percentage points higher (Table 3). Using the regional parameterization algorithm to calculate the parameter $\bar{c}$ gives an even more reliable result, with the values 11–22 percentage points higher in terms of successful nominally predicted PDFs. Hereinafter, we used the regional-oriented parameterization scheme to estimate the future PDFs' parameters of the flood depth of runoff based on the climate change projections.

### 2.3 Data and method application

In performing of the long-term assessment of the extreme flood events in the Russian Arctic the following datasets were used: (i) the climatology for the reference period (Fig. 4 A, B), (ii) the mean values and $C_v$ of the spring flood depth of runoff for the reference period (Fig. 4 C, D), and (iii) the climatology for the projected period (Fig. 4 E, F). The reference climatology was obtained from the catalogs of climatology and the archive of the Arctic and Antarctic Research Institute for 209 meteorological stations (Radionov and Fetterer, 2003; Catalogue, 1989). The climatology was interpolated into the model grid nodes using the algorithm by Hofierka et al. (2002). For the precipitation we use the annual values although the spring floods are formed only by a snowfall and spring rainfall. However, in the Arctic the relationships between spring flood depth of runoff and annual and winter-spring sums of precipitation are similarly strong (Shevnina, 2011).

The mean values and $C_v$ of the spring flood depth of runoff were extracted to the model grid nodes from the maps (Rogdestvenskiy, 1986; Vodogretskiy, 1986). In our study no observations of multi-year runoff were used to evaluate the mean value and $C_v$ for the reference period and no extrapolation was applied for the regions without observations. The climatology for the projected period is provided by the climate models (Pachauri and Reisinger, 2007; Taylor et al., 2012).

In this study, the projections of two Special Report on Emissions Scenarios (SRES: A1B and B1) and two Representative Concentration Pathways (RCPs: 2.6 and 4.5) scenarios were extracted from CMIP3 and CMIP5 data sets. Results of climate models developed by the Max Planck Institute for Meteorology MPIM:ECHAM5 (Roeckner et al., 2003), the Max Planck Institute Earth System Model MPI–ESM (Giorgetta et al., 2013), the Hadley Center for Climate Prediction and Research HadCM3 (Johns et al., 2003), HadGEM2–A (Collins et al., 2008), the Geophysical Fluid Dynamics Laboratory GFDL:CM2 (Delworth et al., 2006) and by the Canadian Center for Climate Modelling Earth System Model CanESM2 (von Salzen et al., 2013) were used. The GCMs used represent the climate projection close to the typical, and show that the hydrological modelling results do not vary much under the climate forcing with the small differences. To obtain the climate forcing, the projected air temperature and precipitation means were corrected using the delta changes method (Fowler et al., 2007). To estimate the future climatology, the relative changes of the variables (in degrees for the temperature and in % for the precipitation) were first calculated based on the historical simulations and observed climatology for the reference period. Then these changes were added/multiplied to the projected climatology.

The corrected mean values of the annual precipitation and annual average air temperature were estimated for the nodes of corresponding climate model grids. Then, for each grid node the means and $C_v$ of the spring flood depth of runoff were extracted from the maps Rogdestvenskiy and Vodogretskiy (1986) with the followed steps: the scanning of the paper maps, the image georeference, the data digitizing and interpolation into the grid nodes

of the particular GCM. The maps were designed based on the observed data for the period since early 1930s till 1980 (Rogdestvenskiy, 1988) which was considered as a reference in our study. In producing these maps the observations on the catchments of medium size (from 1,000 to 50,000 km$^2$) located within the single climate zone were used. Thus, the features of runoff processes on the local scale (appeared on small watersheds) and global scale (revealed on huge watersheds located within several climate zones) as well as floods due to ice jams and tides/surges were not considered.

The values of $\bar{c}$ and $G_{\widetilde{N}}$ were calculated using Eq. 3 for each grid node of the particular climate model. Then, the future first and the second statistical moments from the mean values and $C_v$ of the spring flood depth of runoff were calculated according to Eq. 4 using projected climatology. The values of $C_s$ were estimated using the regional ration of $C_s/C_v$. The maximum discharge with required exceedance probability was calculated according to the Eq. 1 using the projected PDFs of the spring flood depth of runoff, which were modeled using the projected mean value, $C_v$ and $C_s$ (see Section 3 for an example). Our study was performed for the period 2010–2039, since it is within this time interval that the existing and developing socio-economic infrastructure (bridges, oil/gas pipelines, roads and dams) will operate.

## 3. Result and discussion

The analysis of the expected climate change in Russia and particularly over the Arctic region is provided by Govorkova et al. (2008) and Meleshko et al. (2008). These studies include the assessment for the territories of the Russian Federation as a whole. In this study we provide the estimates within the geographical domain of the Russian Arctic, which was outlined according to the hydrological principles as suggested by Ivanov and Yankina (1991) and further used by Nikanorov et al. (2007). For the period of 2010–2039, the climatology averaged over the Russian Arctic is presented in Table 4 for the SRES and RCP scenarios. Generally, an increase of total precipitation over 20 mm (6 %) and warming of over 2.1 °C were predicted according to the SRES scenarios. For the RCP scenarios, the changes of climatology were more pronounced, and the precipitation mean values were expected to increase by more than 40 mm (12 %) and to be accompanied with a warming of 3.3 °C. The strongest increase (over 60 mm or 16 %) in precipitation with the highest warming (over 3.9 °C) was predicted by CaESM2 for the RCP 2.6 scenario (Table 5).

The future means and $C_v$ of the spring flood depth of runoff were assessed from the projected climatology using the method described above. For the entire territory of the Russian Arctic an increase of over 27 mm (17 %) in the mean values and a negligible decrease of $C_v$ were predicted according to the SRES scenarios (Table 4). Using scenarios of the Fifth Assessment Report, the changes in the statistics of the spring flood depth of runoff were more notable: based on the RCP 2.6 scenario, an increase of over 38 mm (23 %) in the mean values and a decrease of over 0.03 (16 %) in the $C_v$'s were expected. The strongest increase (over 45 mm or 27 %) of the means with a lowest decrease of the $C_v$ (over 0.06 or 17 %) was predicted by CaESM2 for the RCP 2.6 scenario.

According to all scenarios considered, the highest increase of the future means of the spring flood depth of runoff (of 30–35 %) was expected for the Arkhangelsk Region and Komi Republic (Fig. 5b). Moderate changes in the mean values (of 10–18 %) are also predicted for Siberia (Fig. 5c and 5d) mostly according to the RCP scenarios. For the

SRES scenarios, an increase of 10–18 % in the mean values was predicted for Kola Peninsula and Karelia (Fig. 5a), accompanied by a decrease of $C_v$.

It is not straightforward to directly compare our results with those of other studies, because we address different flooding characteristics, and only indirect comparison is possible. For the comparison we assume that for the Pearson type III distributions, an increase in the means and $C_v$ leads to an increase of upper-tail values. Subsequently, present 100-year floods will be occur more frequently (Fig. 6). Also, a decrease in the means and $C_v$ leads to a decrease in the upper-tail values. In this case, we can expect that the number of events of 100-year floods decreases. We compared our results with the studies by Hirabayashi et al. (2008; 2013), Lehner et al. (2006) and Dankers and Feyen (2008) using this assumption. For the eastern part of the Arctic, an increase in the historical 100-year maximum discharges is predicted by Hirabayashi et al. (2008; 2013) under the SRES:A1B scenario for the period 2001–2030. This is in accordance with our results; we also expect an increase in the upper-tail runoff values since the mean values and coefficients of variation were estimated to enlarge in average for this region. For the north-east European Arctic we expect a significant increase in the frequency of present 100-year flood events. This is in contrast to Hirabayashi et al. (2013), which study presents the global scale estimates of the projected change in the flood frequency. The flood frequency is decreased in many regions of northern and eastern Europe according to Hirabayashi et al. (2013). The feasible reason for such disagreement is the spatial coarseness of the model used by Hirabayashi et al. (2013), which was calibrated using observations from watersheds larger than 100,000 km$^2$. In our study, the probabilistic model was calibrated using observations for watersheds of medium range. Lehner et al. (2006) used the WaterGAP model with climate projections derived from the HadCM3 and ECHAM4/OPYC3 GCMs. The results suggest that present 100-year flood events will occur more frequently in the north-eastern European Arctic in 2020s, which is in accordance with our results.

For Kola Peninsula and Karelia, we predicted a decrease of the mean values with slight increase of the $C_v$ according to the SRES:A1B and SRES:B1 scenarios. Dankers and Feyen (2008) suggested a strong decrease of present 100-year floods for north-eastern Europe (i.e. Finland, northern Russia and part of the Baltic States) under the SRES:A2 and SRES:B2 scenarios, which is in general agreement with our results. A similar tendency for the predicted maximal discharges to decrease was obtained for the northern Finland (Veijalainen et al., 2010).

There are several sources of uncertainties in the method described above: (1) from the assumed (given a priori) type of distribution (Pearson type III); (2) from the limited length of hydrological time series, which were used to evaluate the parameters of the distribution for the reference period; (3) from the limited length of meteorological time series to evaluate the climatology for the model's parameterization; (4) from the uncertainties in future climatology provided by climate models (forcing); (5) from the mapping errors due to interpolation techniques; (6) from the errors due to the calculation of the maximal discharges from the spring flood depth of runoff (Eq. 1). The uncertainties inherent to the simulated PDFs' parameters include items 1–5 from the list above. These uncertainties are evaluated by Kovalenko (1993) for the maps of means/$C_v$, provided by Pogdestvenskiy (1986) and Vodogretskiy (1986) in assumption that the errors in the future and past climatology are the same. The average percentage errors in the projected means/$C_v$ are equals to 15 % / 25 %, thus it is suggested to consider the changes in the PDFs' parameters to be substantial if they exceed the reference values for more than these thresholds. Then the regions with substantial changes in the means and $C_v$ of the spring flood flow depth were outlined (Fig. 7).

In these regions the frequency and magnitude of floods were expected to differ substantially from the historical (reference) period. The changes in the mean values and coefficient of variation were predicted relying on the outputs of the climate models of the Max Planck Institute for Meteorology: MPIM:ECHAM5 for the SRES:B1 scenario and MPI-ESM-LR for the RCP 2.6 scenario. A substantial increase in the mean values is expected for the Arkhangelsk region, Komi Republic and Eastern Siberia (see Fig. 8 for the boundary of the regions). These are warning regions where the flood related risks for hydraulic constructions in the future may be different from the past. In these regions, the calculations of the maximal discharges should be corrected in line with the expected climate change.

The example of the climate-based correction for the Nadym River at Nadym City according to climate model outputs for the RCP 2.6 scenario is given below. A new bridge over the Nadym River is currently in planning. To assess the bridge height and cost the maximal discharge of rare occurrence (e.g. 1% exceedance probability) is required. The watershed of the Nadym River is located in the region, where an increase of the means of the spring flood depth of runoff was predicted under RCP2.6 scenario (Fig. 7, right, upper plot). Thus, the climate change impacted upper-tail maximal discharge may well be larger than the value estimated from the observed time series. Hydrological observations for the Nadym River are available at the Nadym City (gauge number 11805 in the bottom panel of Fig. 8, the watershed area is 48,000 km$^2$). The statistics of the spring flood depth of runoff for this gauge were calculated from the observations for the period of 1954–1980, which was considered as reference in this case (Table 6). The reference climatology was calculated by averaging the observations from the regular meteorological sites for the Nadym River catchment area for the same period. Then, the delta corrected projected climatology for the period of 2010–2039 under the RCP 2.6 scenario was obtained from the CMIP5 dataset. The parameter $G_{\widetilde{N}}$ was estimated according to the observed climatology and the parameter $\overline{c}$ was calculated based on the projected climatology according to Shevnina (2012). These values were used to predict the first and second statistical moments, and coefficient of variation ($m_1$, $m_2$ and $C_v$) of the spring flood depth of runoff. The projected $C_s$ was estimated from the given ratio of $C_s/C_v$. The projected PDF was obtained from these values, and the spring flood depth of runoff 1 % exceedance probability ($h_{1\%}$, mm) was calculated (Table 6). The confidence intervals for the reference values of $h_{1\%}$ were calculated using the formulas suggested by Ashkar and Bobée (1988) in assumption that the given distribution is Pearson type III. The 90% confidence interval for the reference $h_{1\%}$ equal to ±64.5 mm, which is about 23 % of the quantile value. The projected values of $h_{1\%}$ are within these uncertainties for all considering climate scenarios (Table 6), thus due to the short time series we can't prove that the uncertainties due to the short time series make the future changes in $h_{1\%}$ in the future are statistically insignificant. However, we suggest to take into account for the projected climatology in calculation of hydrological risks because of practical reasons: it is better to prevent an accident rather than to deal with its consequences, which may be more expensive than the initial investment (Räisänen and Palmer, 2001). Finally, the maximal discharge with 1 % exceedance probability ($Q_{1\%}$, m$^3$s$^{-1}$) was estimated from $h_{1\%}$ according to Eq. 1. The values of the parameters of Eq. 1 were obtained from the look-up tables (Guideline, 1984): the value of $k_0$ was considered to be constant for the reference and projected periods and it was set to be equal to 1 in our example for sake of simplicity; $\mu$ equals to 1.0; $\delta, \delta_1, \delta_2$ equal to 0.84, 0.06 and 0.08 correspondingly; $b$ equals to 1.0 (km$^2$) and $n$ equals to 0.17.

For the period 2010–2039 the maximal discharge of 1% exceedance probability, calculated with averaging of the multi-model output, is 570 $m^3s^{-1}$ (this is the difference between the historical and average projected discharges) larger than the discharge of the same probability of exceedance calculated from the observations. The largest increase of the maximal discharge was predicted according to the CanESM2 model (over 7% larger than the historical value). The

maximal discharge of 8572 $m^3s^{-1}$ changed the probability of exceedance from 1% (calculated from the observations) to 2.5 % (calculated according to the averaged climate projections).

**4. Conclusions**

A probabilistic model was applied in estimating the impact of the climate changes to the frequency and magnitude of extreme floods in the Russian Arctic. The probabilistic hydrological model allows the calculation

of the future runoff extremes with the required exceedance probability without a need to simulate the future runoff time series. The projected meteorological mean values for the periods of 20–30 years were used to estimate the future means, $C_v$ and $C_s$ of the spring flood depth of runoff, and to model the PDFs with a Pearson type III distribution. The future frequency and magnitude of extreme floods with a required exceedance probability were then evaluated from the simulated PDFs.

In this study, to perform the model cross-validation the runoff data were extracted from the official issues of Roshydromet, however, in calculating multi-year time series of spring flood depth of runoff (and maximal discharge), the global and regional runoff databases may be also used since daily discharge time series are required. The examples of the datasets are (i) the Global Runoff Data Centre, Germany; (ii) the Environmental Information System (HERTTA), Finnish Environment Institute; Vattenwebb by the Swedish Meteorological and

Hydrological Institute. For the other regions, in performing the regional scale assessments the steps are following: (i) to choose the middle size watersheds with catchment area from 1,000 to 50,000 $km^2$; (ii) to calculated the multi-year time series of runoff (yearly maximal discharges or flood depth of runoff) from the daily runoff time series; (iii) to select the time period without statistically significant trends (reference period); (iv) to estimate the mean values, $C_v$ and $C_s$ from the observed time series of runoff or to evaluate them from the

regional maps (i.e. Spence and Burke, 2008) as well as the statistics of the precipitation and air temperature; (v) to perform the model parameterization using general (Kovalenko et al., 2010) or the regional-oriented schemes (Shevnina, 2012); (vi) to assess the mean values of the precipitation and air temperature from the results of the GCM/RCM models for the future; and (vii) to evaluate the future means, $C_v$ and $C_s$ of multi-year runoff with Eq. (4). To perform the model cross-validation and to develop the regional-oriented parameterization scheme, the

multi-year time series of runoff with the periods of statistically significant shifts in the mean values and $C_v$ are required.

The probabilistic model was further applied for a regional scale assessment of extreme flood events for the Russian Arctic. The regional-oriented parameterization by Shevnina (2012) allows a successful prediction of 67–83 % of the PDFs (see Section 2.2). The projected mean values, $C_v$ and $C_s$ of the spring flood depth of runoff

for the period 2010–2039 were estimated under the SRES:A1B, SRES:B1, RCP 2.6 and RCP 4.5 climate scenarios with outputs of three climate models. For the region studied, an increase of 17–23 % in the mean values of

spring flood depth of runoff and a decrease of 5–16 % in the $C_v$ were predicted depending on the scenarios considered. For the northwest of the Russian Arctic, an increase in the means and a decrease of the $C_v$ were predicted. The regions with substantial changes in the mean values (over 15 %) and $C_v$ (over 25 %) were defined for 2010–2039. For the territories where the means and $C_v$ increased substantially, the extreme floods are projected to occur more frequently and the risk of flooding is increased. We suggest to correct the hydrological engineering calculations, and to account for the projected climatology. It might reduce the risk of a potential hazard for the hydraulic constructions, the oil-gas industry, the transport infrastructure and population located in these warning regions.

The model presented in this study provides an affordable method to produce forecasts of extreme flood events (in form of PDF or as maximal discharge with a required exceedance probability) under the projected climate change. This is possible due to low number of the simulated variables and parameters. The regionally-oriented parameterization of the model is also relatively simple and may be improved by involving a variance of precipitation, which could be obtained from the projected climatology (Meehl et al., 2011). However, due to its various simplifications, the model presented in this study does not allow an estimation of possible changes in spring flood timing or changes of intra-seasonal runoff variability for a particular watershed. On a regional scale, however, the method presented provides an explicit advantage to estimate extreme hydrological events under altered climate, especially for regions with an insufficient observational data. It could be useful for a broad-scale assessment to define the warning regions, where crucial increase/decrease of the extreme flood events is expected. When the warning regions are defined, a catchment-scale rainfall-runoff model could be applied to further distinguish details not anticipated by the method described in this study. Such models also allow to evaluate the value of the spring flood coincidence factor $k_0$ (Eq. 1) for the projected periods (which was constant in our calculations). The evaluation and inter-comparison of the presented model and rainfall-runoff models is of high interest.

Another weak point of the method is the use of look-up tables for physiographic parameters. In our study, to calculate the extreme discharges of the Nadym River we used look-up tables for the territory of the former Soviet Union from Guideline (1984). For other regions world-wide, these physiographic parameters may be derived from spatially distributed datasets, e.g. according to Bertholomee and Belward (2005). Also, an issue to be studied is the effect of the spatial resolution of projected climatology on the ability of the this model to estimate the frequency/magnitude of extreme floods for watersheds of different size.

The method described in this study was simplified for the use of engineering calculations, as the projected climatology for the periods of 20–30 years recommended by IPCC (Pachauri and Reisinger, 2007) assumes a quasi-stationary climate. In general, the quasi-stationarity assumption may be eliminated and a non-stationary regime could be considered. In this case, the PDFs could be evaluated based on the full form of the Fokker–Planck–Kolmogorov equation (Domínguez and Rivera, 2010) with the multi-model climate ensemble approach (Tebaldi and Knutti, 2007).

**5 Annex**

The concept of the probabilistic modeling to perform a hydrological response to an expected climate change was proposed by Kovalenko (1993), it is presented further as provided in Kovalenko et al. (2010). This approach considers multi-year runoff time series (annual, maximal and minimal) as realizations of a discrete stochastic process presented as a Markov chain (Rogdestvenskiy, 1988). Then, a first order ordinary differential equation is used as a lump hydrological model to perform multi-year flow time series:

$$dQ/dt = -(1/k\tau)Q + \dot{X}/\tau \text{, (A1)}$$

where $Q$ is some runoff characteristic depending on a task (the discharge, the volume per year, the runoff depth per year, *etc.* – "model output"); $\dot{X}$ is the precipitation amount per year ("model input"); $k$ is the runoff coefficient; $\tau$ is the time of reaction of the watershed to the incoming precipitation (here, $\tau = 1$ year, which physically means that the precipitation amount during one year generate the runoff from the watershed during one year); $t$ is the time interval, equals to one year. Denoting $c = 1/k\tau$ and $N = \dot{X}/\tau$ and adding random components ($\widetilde{c}$, $\widetilde{N}$ are performed as "white noise") to $c = \bar{c} + \widetilde{c}$ and $N = \bar{N} + \widetilde{N}$ we obtain the stochastic differential equation:

$$dQ = [-(\bar{c} + \widetilde{c})Q + (\bar{N} + \widetilde{N})]dt \text{ . (A2)}$$

The random components are mutually correlated.

The solution of Eq. A2 is statistically equivalent to the solution of the Fokker-Planck-Kolmogorov equation:

$$\frac{\partial p(Q;t)}{\partial t} = -\frac{\partial}{\partial Q}(A(Q;t)p(Q;t)) + 0.5\frac{\partial^2}{\partial Q^2}(B(Q;t)p(Q;t)) \text{, (A3)}$$

where $p(Q;t)$ is the probability density function of the multi-year runoff characteristic ($Q$ is considered now as a random value); $A(Q;t)$ and $B(Q;t)$ are the drifting and diffusion coefficients:

$$A(Q;t) = -(\bar{c} + 0.5G_{\widetilde{c}})Q - 0.5G_{\widetilde{c}\widetilde{N}} + \bar{N} \text{ ;}$$

$$B(Q;t) = G_{\widetilde{c}}Q^2 - 2G_{\widetilde{c}\widetilde{N}} + G_{\widetilde{N}} \text{ , (A4)}$$

here, $G_{\widetilde{c}}$ and $G_{\widetilde{N}}$ are the measures of variability of $c$ and $N$; $G_{\widetilde{c}\widetilde{N}}$ is the measure of correlation between the variability of $G_{\widetilde{c}}$ and $G_{\widetilde{N}}$ .

In engineering hydrological applications and flood frequency analysis only three-parametric probability density functions are used (Bulletin 17–B, 1988). Then Eq. A3 may be simplified to a system of ordinary differential equations for three statistical moments $m_i$ ($i = 1, 2, 3$):

$$dm_1/dt = -(\bar{c} - 0.5G_{\widetilde{c}})m_1 - 0.5G_{\widetilde{c}\widetilde{N}} + \bar{N} \text{ ;}$$
$$dm_2/dt = -2(\bar{c} - G_{\widetilde{c}})m_2 + 2\bar{N}m_1 - 3G_{\widetilde{c}\widetilde{N}}m_1 + G_{\widetilde{N}} \text{ ;} \qquad \text{(A5)}$$
$$dm_3/dt = -3(\bar{c} - 1.5G_{\widetilde{c}})m_3 + 3\bar{N}m_2 - 7.5G_{\widetilde{c}\widetilde{N}}m_2 + 3G_{\widetilde{N}}m_1 \text{ .}$$

This system can be used to calculate the statistics of the multi-year runoff: the mean $\bar{Q} = m_1$, the coefficient of variation $C_v = \sqrt{(m_2 - m_1^2)}/m_1$ and the coefficient of skewness $C_s = (m_3 - 3m_2m_1 + 2m_1^3)/(C_v^3 m_1^3)$ .

Additionally, the constant value of $C_s/C_v$ ratio for the projected time period was used to simplify the Eq. A5, it is commonly applied in engineering hydrological applications to estimate the regional $C_s$. Also, the climate scenarios are distributed by IPCC as mean values of meteorological variables for the periods of 20–30 years. Thus, scenarios are presented expected for climate changes with an assumption of "quasi-stationarity" and this

may also be applied for the hydrological regime. This allows further simplifications of Eq. A5: $dm_i/dt \approx 0$ and

$G_{\widetilde{c}}, G_{\widetilde{c}\,\widetilde{N}} = 0$ within these periods. Hence, Eq. A5 may be reduced to only two algebraic equations for $m_1$ and

$m_2$:

$$-\bar{c}\, m_1 + \bar{N} = 0$$
$$-2\bar{c}\, m_2 + 2\bar{N}\, m_1 + G_{\widetilde{N}} = 0$$ .

This system may be applied to estimate the multi-year hydrological statistical moments directly from

climatology for each "quasi-stationary" time period (e.g. 2010–39).

*Acknowledgements.* This study was funded through the Ministry of Education and Science of the Russian Federation (project 1413) and supported by the Academy of Finland (contract 283101). The authors are very thankful to the Reviewers of BER and HESS, who provided very useful comments and suggestion in improving

of the manuscript. Our special thanks to Lynn.

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

**Table 1.** The multi-year statistics of the spring flood depth of runoff and the climatology for the sub-periods with the statistically significant shift in the mean values of spring flood depth of runoff.

**Notations:** $m_1$ and $m_2$ are the first and second statistical moments of the spring flood depth of runoff; $C_v$ is the coefficient of variation; $C_s$ is the coefficient of skewness; $\bar{N}$ is the mean values of annual precipitation amount; $\bar{T}$ is the mean values of annual air temperature.

| Gauge ID | River | Catchment area [km²] | Period | $m_1$ [mm] | $m_2$ [mm²] | $C_v$ | $C_s/C_v$ | $\bar{N}$ [mm] | $\bar{T}$ [°C] |
|---|---|---|---|---|---|---|---|---|---|
| 01176 | Bohapcha | 13600 | 1934–1949 | 111 | 15401 | 0.50 | 2.5 | 421 | −12.1 |
| | | | 1950–1980 | 141 | 23907 | 0.45 | 2.8 | 435 | −12.4 |
| 01309 | Seimchan | 2920 | 1941–1956 | 190 | 40779 | 0.36 | 3.1 | 373 | −11.5 |
| | | | 1957–1977 | 157 | 25842 | 0.22 | 5.1 | 305 | −11.4 |
| 01623 | Srednekan | 1730 | 1935–1950 | 148 | 25067 | 0.38 | 4.0 | 426 | −10.7 |
| | | | 1951–1980 | 180 | 36145 | 0.34 | 4.5 | 431 | −11.1 |
| 03403 | Malaya Kuonapka | 2030 | 1943–1985 | 97.5 | 10848 | 0.36 | 0.8 | 255 | −13.8 |
| | | | 1986–2002 | 116 | 14297 | 0.25 | 1.1 | 262 | −13.1 |
| 03414 | Yana | 45300 | 1935–1964 | 41.1 | 2190 | 0.55 | 1.2 | 177 | −14.8 |
| | | | 1965–2002 | 52.1 | 3456 | 0.48 | 1.4 | 178 | −14.6 |
| 03518 | Nera | 2230 | 1944–1985 | 67.0 | 5439 | 0.46 | 0.8 | 227 | −15.8 |
| | | | 1986–2002 | 84.6 | 8214 | 0.37 | 1.0 | 222 | −14.4 |
| 09425 | Turukhan | 10100 | 1941–1970 | 232 | 56198 | 0.21 | 1.3 | 491 | −7.4 |
| | | | 1971–1999 | 260 | 70304 | 0.20 | 1.4 | 494 | −7.4 |
| 11574 | Pyakupur | 31400 | 1954–1970 | 142 | 21140 | 0.22 | 4.2 | 482 | −6.4 |
| | | | 1971–2001 | 162 | 27884 | 0.23 | 3.7 | 514 | −6.0 |
| 11805 | Nadym | 48000 | 1955–1974 | 162 | 27632 | 0.23 | 3.0 | 490 | −6.4 |
| | | | 1975–1991 | 140 | 21607 | 0.32 | 2.2 | 471 | −5.0 |
| 70047 | Solza | 1190 | 1928–1958 | 190 | 38356 | 0.25 | 0.9 | 525 | 1.3 |
| | | | 1959–1980 | 155 | 26046 | 0.29 | 0.8 | 552 | 1.0 |
| 70153 | Yug | 15200 | 1931–1946 | 126 | 16716 | 0.23 | 2.0 | 575 | 1.6 |
| | | | 1947–1980 | 144 | 22994 | 0.33 | 1.4 | 591 | 1.6 |
| 70180 | Vychegda | 26500 | 1930–1956 | 147 | 22960 | 0.25 | 0.0 | 491 | −0.1 |
| | | | 1957–1980 | 167 | 29632 | 0.25 | 0.0 | 550 | −0.5 |
| 70360 | Lodma | 1400 | 1939–1958 | 219 | 53184 | 0.33 | 1.2 | 533 | 0.7 |
| | | | 1959–1977 | 174 | 32650 | 0.28 | 1.4 | 546 | 0.7 |
| 70366 | Kuloy | 3040 | 1927–1958 | 134 | 20549 | 0.38 | 1.4 | 467 | 1.0 |
| | | | 1959–1980 | 110 | 13582 | 0.35 | 1.5 | 446 | 0.6 |
| 70410 | Pechora | 9620 | 1914–1930 | 302 | 94159 | 0.18 | -0.4 | 516 | −1.0 |
| | | | 1931–1993 | 276 | 79535 | 0.21 | -0.3 | 564 | −1.0 |
| 70414 | Pechora | 29400 | 1938–1956 | 250 | 65806 | 0.23 | 0.5 | 490 | −1.0 |

| Gauge ID | River | Catchment area [km²] | Period | $m_1$ [mm] | $m_2$ [mm²] | $C_v$ | $C_s/C_v$ | $\bar{N}$ [mm] | $\bar{T}$ [°C] |
|---|---|---|---|---|---|---|---|---|---|
| | | | 1957–1980 | 278 | 79262 | 0.16 | 0.8 | 601 | −1.3 |
| 70466 | Usa | 2750 | 1936–1957 | 385 | 155399 | 0.22 | 1.5 | 483 | −4.3 |
| | | | 1958–1980 | 424 | 185601 | 0.18 | 1.8 | 558 | −5.3 |
| 70509 | Izhma | 15000 | 1933–1949 | 189 | 37779 | 0.24 | 0.1 | 465 | −0.5 |
| | | | 1950–1980 | 160 | 26839 | 0.22 | 0.1 | 534 | −0.9 |
| 70522 | Ukhta | 4290 | 1934–1949 | 170 | 30706 | 0.25 | 0.9 | 473 | −0.5 |
| | | | 1950–1980 | 144 | 22032 | 0.25 | 0.9 | 535 | −0.5 |
| 70531 | Pizhma | 4890 | 1937–1964 | 129 | 18041 | 0.29 | 0.9 | 486 | −1.7 |
| | | | 1965–1980 | 150 | 24264 | 0.28 | 0.9 | 552 | −2.3 |
| 71104 | Kola | 3780 | 1928–1958 | 182 | 35539 | 0.27 | 2.6 | 350 | 0.5 |
| | | | 1959–1994 | 203 | 43785 | 0.25 | 2.6 | 459 | 0.1 |
| 71199 | Umba | 6920 | 1931–1958 | 180 | 34762 | 0.27 | 0.6 | 414 | −1.1 |
| | | | 1959–1994 | 149 | 23942 | 0.28 | 0.6 | 475 | −1.6 |
| 71241 | Yena | 1600 | 1934–1948 | 100 | 10625 | 0.25 | 0.7 | 451 | 0.2 |
| | | | 1949–1980 | 129 | 18041 | 0.29 | 0.7 | 557 | −0.3 |

**Table 2.** The model parameters and the nominally predicted multi-year statistics of the spring flood depth of runoff for the catchments selected for the cross-validation.

**Notations:** $m_{1f}$ and $m_{2f}$ are the nominally predicted first and second statistical moments of the spring flood depth of runoff; $C_v$ is the nominally predicted coefficient of variation; $C_s$ is the nominally predicted coefficient of skewness; $\bar{c}_f$ is the inverse of the runoff coefficient times the watershed reaction delay; $G_{\widetilde{N}f}$ characterizes the variability of the annual precipitation amount.

| Gauge ID | Lat/Lon | Period | $G_{\widetilde{N}f}$ [mm²] | $\bar{c}_f$ | $m_{1f}$ [mm] | $m_{2f}$ [mm²] | $C_{vf}$ | $C_{sf}$ |
|---|---|---|---|---|---|---|---|---|
| 01176 | 62°06′N / 150°37′E | 1934–1949 | 23366 | 3.79 | 115 | 16234 | 0.48 | 1.20 |
| | | 1950–1980 | 24841 | 3.09 | 136 | 22647 | 0.46 | 1.28 |
| 01309 | 63°17′N / 152°02′E | 1941–1956 | 18370 | 1.96 | 155 | 28815 | 0.44 | 1.38 |
| | | 1957–1977 | 4635 | 1.94 | 141 | 20941 | 0.25 | 1.26 |
| 01623 | 62°22′N / 152°20′E | 1935–1950 | 18208 | 2.88 | 150 | 25584 | 0.38 | 1.50 |
| | | 1951–1980 | 17936 | 2.39 | 178 | 35398 | 0.34 | 1.54 |
| 03403 | 70°11′N / 113°57′E | 1943–1985 | 6477 | 2.60 | 101 | 11383 | 0.35 | 0.27 |
| | | 1986–2002 | 3799 | 2.26 | 113 | 13587 | 0.26 | 0.29 |
| 03414 | 67°24′N / 137°15′E | 1935–1964 | 4390 | 4.32 | 42.0 | 2209 | 0.55 | 0.68 |
| | | 1965–2002 | 4347 | 3.36 | 52.7 | 3425 | 0.48 | 0.68 |
| 03518 | 64°43′N / 144°37′E | 1944–1985 | 6436 | 3.39 | 66.0 | 5243 | 0.47 | 0.38 |
| | | 1986–2002 | 5167 | 2.61 | 86.9 | 8543 | 0.36 | 0.36 |
| 09425 | 65°58′N / 84°17′E | 1941–1970 | 10047 | 2.12 | 233 | 56857 | 0.21 | 0.27 |
| | | 1971–1999 | 10275 | 1.90 | 258 | 69485 | 0.20 | 0.27 |
| 11574 | 64°56′N / 77°48′E | 1954–1970 | 6625 | 3.39 | 151 | 23906 | 0.21 | 0.86 |
| | | 1971–2001 | 10408 | 3.17 | 152 | 24718 | 0.27 | 0.27 |
| 11805 | 65°39′N / 72°42′E | 1955–1974 | 8398 | 3.02 | 156 | 25636 | 0.24 | 0.72 |
| | | 1975–1991 | 13505 | 3.36 | 146 | 23220 | 0.31 | 0.66 |
| 70047 | 64°41′N / 39°32′E | 1928–1958 | 12469 | 2.76 | 200 | 42164 | 0.24 | 0.21 |
| | | 1959–1980 | 14391 | 3.56 | 147 | 23753 | 0.30 | 0.23 |
| 70153 | 60°12′N / 47°00′E | 1931–1946 | 7665 | 4.56 | 130 | 17612 | 0.22 | 0.46 |
| | | 1947–1980 | 18536 | 4.10 | 140 | 21886 | 0.34 | 0.48 |
| 70180 | 61°52′N / 53°49′E | 1930–1956 | 9022 | 3.34 | 165 | 28465 | 0.22 | –0.01 |
| | | 1957–1980 | 11481 | 3.29 | 149 | 23969 | 0.28 | –0.01 |
| 70360 | 64°25′N / 41°03′E | 1939–1958 | 25423 | 2.43 | 224 | 55552 | 0.32 | 0.38 |
| | | 1959–1977 | 14897 | 3.14 | 170 | 31225 | 0.29 | 0.40 |
| 70366 | 64°59′N / 43°42′E | 1927–1958 | 18073 | 3.49 | 128 | 18970 | 0.40 | 0.55 |
| | | 1959–1980 | 12020 | 4.05 | 115 | 14749 | 0.33 | 0.51 |
| 70410 | 61°52′N / 56°57′E | 1914–1930 | 10098 | 1.71 | 330 | 111916 | 0.16 | –0.06 |
| | | 1931–1993 | 13730 | 2.04 | 253 | 67121 | 0.23 | –0.08 |
| 70414 | 62°57′N / 56°56′E | 1938–1956 | 12960 | 1.96 | 307 | 97330 | 0.19 | 0.10 |
| | | 1957–1980 | 8554 | 2.16 | 227 | 53351 | 0.20 | 0.15 |

| Gauge ID | Lat/Lon | Period | $G_{\widetilde{N}f}$ [mm$^2$] | $\bar{c}_f$ | $m_{1f}$ [mm] | $m_{2f}$ [mm$^2$] | $C_{vf}$ | $C_{sf}$ |
|---|---|---|---|---|---|---|---|---|
| 70466 | 66°36′N / 60°52′E | 1936–1957 | 18000 | 1.25 | 445 | 205006 | 0.19 | 0.29 |
|  |  | 1958–1980 | 15331 | 1.32 | 367 | 140521 | 0.21 | 0.38 |
| 70509 | 63°49′N / 53°58′E | 1933–1949 | 10124 | 2.46 | 217 | 49166 | 0.21 | 0.03 |
|  |  | 1950–1980 | 8271 | 3.34 | 139 | 20651 | 0.25 | 0.03 |
| 70522 | 63°35′N / 53°51′E | 1934–1949 | 10051 | 2.78 | 192 | 38779 | 0.22 | 0.19 |
|  |  | 1950–1980 | 9630 | 3.72 | 127 | 17504 | 0.28 | 0.25 |
| 70531 | 65°17′N / 51°55′E | 1937–1964 | 10545 | 3.77 | 147 | 22867 | 0.26 | 0.23 |
|  |  | 1965–1980 | 12983 | 3.68 | 132 | 10205 | 0.32 | 0.30 |
| 71104 | 68°56′N / 30°55′E | 1928–1958 | 9287 | 1.92 | 239 | 59383 | 0.21 | 0.13 |
|  |  | 1959–1994 | 11647 | 2.26 | 155 | 26536 | 0.33 | 0.85 |
| 71199 | 66°52′N / 33°20′E | 1931–1958 | 10865 | 2.30 | 207 | 45013 | 0.24 | 0.15 |
|  |  | 1959–1994 | 11098 | 3.19 | 130 | 18606 | 0.32 | 0.24 |
| 71241 | 67°18′N / 32°08′E | 1934–1948 | 5638 | 4.51 | 124 | 15878 | 0.20 | 0.53 |
|  |  | 1949–1980 | 12086 | 4.32 | 104 | 1209 | 0.36 | 0.26 |

**Table 3.** The percentage of successful fits between the nominally predicted and empirical PDFs according to the goodness-of-fit tests for 0.05 level of statistical significance.

| Version of the nominal prediction | Kolmogorov-Smirnov one-sample test | Pearson chi-squared test |
|---|---|---|
| No model | 63 | 41 |
| Model with parameterization by Kovalenko et al. (2010) | 67 | 51 |
| Model with regional-oriented parameterization by Shevnina (2012) | 74 | 63 |

**Table 4.** The reference (1930–1980) and projected climatology (2010–2039) and statistics of the spring flood flow depth of runoff averaged for the entire territory of the Russian Arctic.

| Multi-year statistical values | Reference climatology | Fourth Assessment Report (AR4) | | Fifth Assessment Report (AR5) | |
|---|---|---|---|---|---|
| | | SRES:A1B | SRES:B1 | RCP 4.5 | RCP 2.6 |
| The annual amount of precipitation mean value ($\bar{N}$ mm) | 378 | 400 | 402 | 424 | 424 |
| The average annual air temperature mean value ($\bar{T}$ °C) | −10.3 | −8.2 | −8.2 | −6.9 | −7.2 |
| The spring flood depth of runoff mean value ($m_1$ mm) | 162 | 189 | 190 | 201 | 199 |
| The coefficient of variation of the spring flood depth of runoff ($C_v$) | 0.30 | 0.30 | 0.29 | 0.29 | 0.25 |

**Table 5.** Projected (2010–2039) climatology and statistics of the spring flood depth of runoff averaged for the entire territory of the Russian Arctic according to the results of different climate models.

**Notations:** $\bar{N}$ is the mean values of annual precipitation amount; $\bar{T}$ is the mean values of annual air temperature; $m_1$ is the mean value of the spring flood depth of runoff; $C_v$ is the coefficient of variation of the spring flood depth of runoff.

| Dataset | Scenario | GCM | $\bar{N}$, [mm] | $\bar{T}$, [°C] | $m_1$, [mm] | $C_v$ |
|---------|----------|-----|-----------------|------------------|-------------|-------|
| AR4 | SRES:A1B | MPIM:ECHAM5 | 393 | −8.6 | 184 | 0.30 |
| | | HadCM3 | 403 | −7.9 | 191 | 0.30 |
| | | GFDL:CM2 | 404 | −8.2 | 192 | 0.29 |
| | SRES:B1 | MPIM:ECHAM5 | 385 | −8.4 | 182 | 0.30 |
| | | HadCM3 | 405 | −8.1 | 191 | 0.30 |
| | | GFDL:CM2 | 415 | −8.2 | 196 | 0.28 |
| AR5 | RCP4.5 | MPI–ESM | 421 | −6.9 | 201 | 0.26 |
| | | HadGEM2–A | 420 | −7.0 | 199 | 0.26 |
| | | CanESM2 | 436 | −6.7 | 204 | 0.25 |
| | RCP2.6 | MPI–ESM | 415 | −7.2 | 197 | 0.26 |
| | | HadGEM2–A | 419 | −7.9 | 194 | 0.26 |
| | | CanESM2 | 438 | −6.4 | 207 | 0.24 |

**Table 6.** Climatology and the statistics of the extreme flood runoff for the Nadym River at the Nadym City evaluated from the observations and under the climate projection RCP 2.6 for the period 2010–2039.

**Notations:** $\bar{N}$ is the mean values of annual precipitation amount; $\bar{T}$ is the mean values of annual air temperature; $m_1$ is the mean value of the spring flood depth of runoff; $C_v$ is the coefficient of variation of the spring flood depth of runoff, $h_{1\%}$ is the spring flood depth of runoff with exeedance of 1%, $Q_{1\%}$ is the maximal discharge with exeedance of 1%.

| Multi-year values | Period of 1954–1980 | Result according to GCM | | | |
|---|---|---|---|---|---|
| | | HadGEM2-A | MPI-ESM-LR | CanESM2 | Multi model |
| $\bar{N}$ mm | 431 | 483 | 491 | 519 | 498 |
| $\bar{T}$ °C | −5.9 | −4.0 | −2.9 | −2.4 | −3.1 |
| $m_1$ mm | 160 | 180 | 184 | 197 | 187 |
| $C_v$ | 0.28 | 0.25 | 0.23 | 0.19 | 0.22 |
| $h_{1\%}$ mm | 277 | 297 | 293 | 297 | 296 |
| $Q_{1\%}$ m$^3$s$^{-1}$ | 8572 | 9177 | 9062 | 9191 | 9144 |

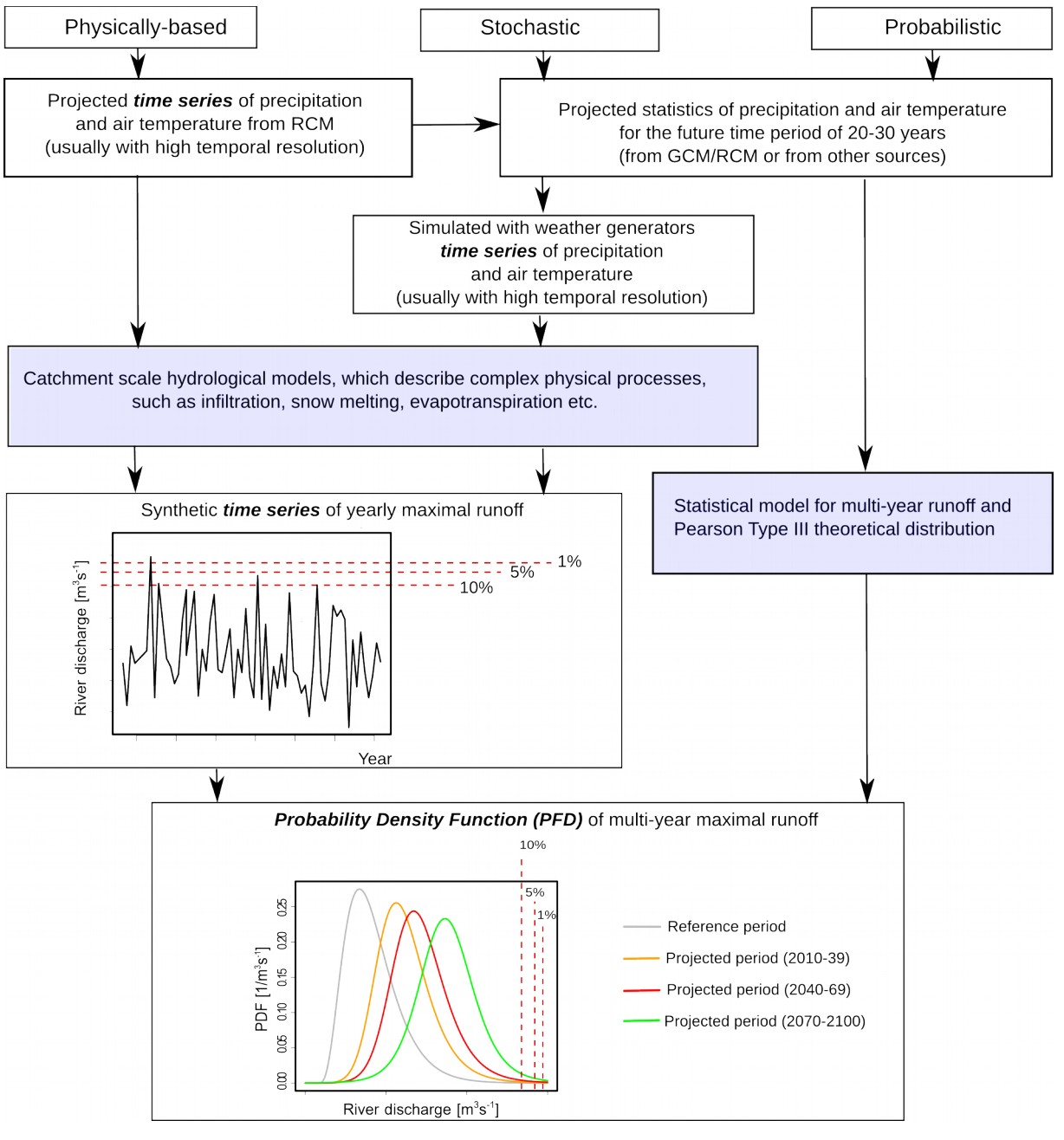

**Figure 1.** Three approaches to evaluate a hydrological response to the expected climate change.

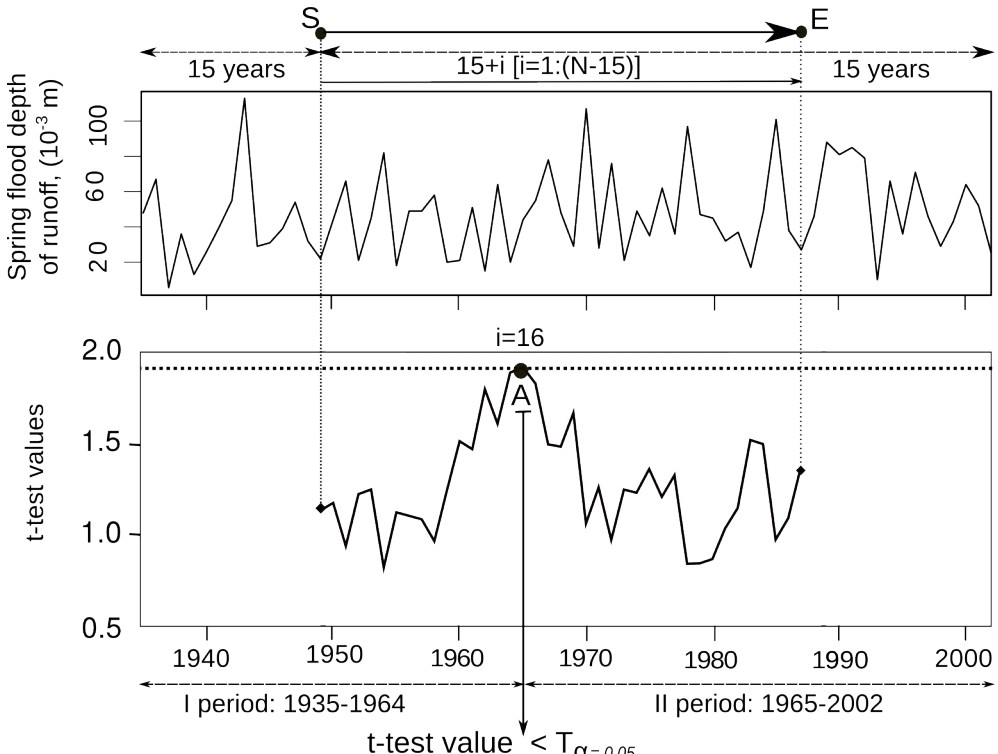

**Figure 2.** The partition of the observed time series of the spring flood depth of runoff (top) into sub-periods with statistically significant shift in the mean value by Student's *t*-test (bottom) for the Yana River at the Verkhoyansk City gauge: $T_{\alpha=0.05}$ is critical value of the *t*-test at the threshold of the statistical significance equal to 0.05 (dotted line on bottom). See the text for explanation of A, S and E.

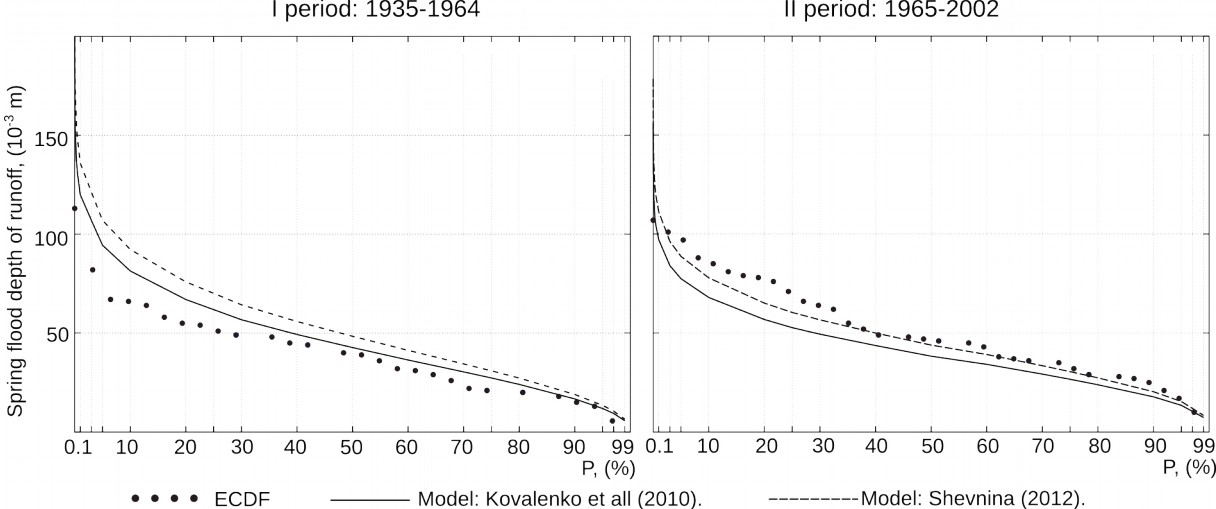

**Figure 3.** The nominally predicted exceedance probability curves fitted to the empirical data for the sub-periods with statistically significant shift in the mean value: the Yana River at the Verkhoyansk City (ECDF – empirical exceedance probability).

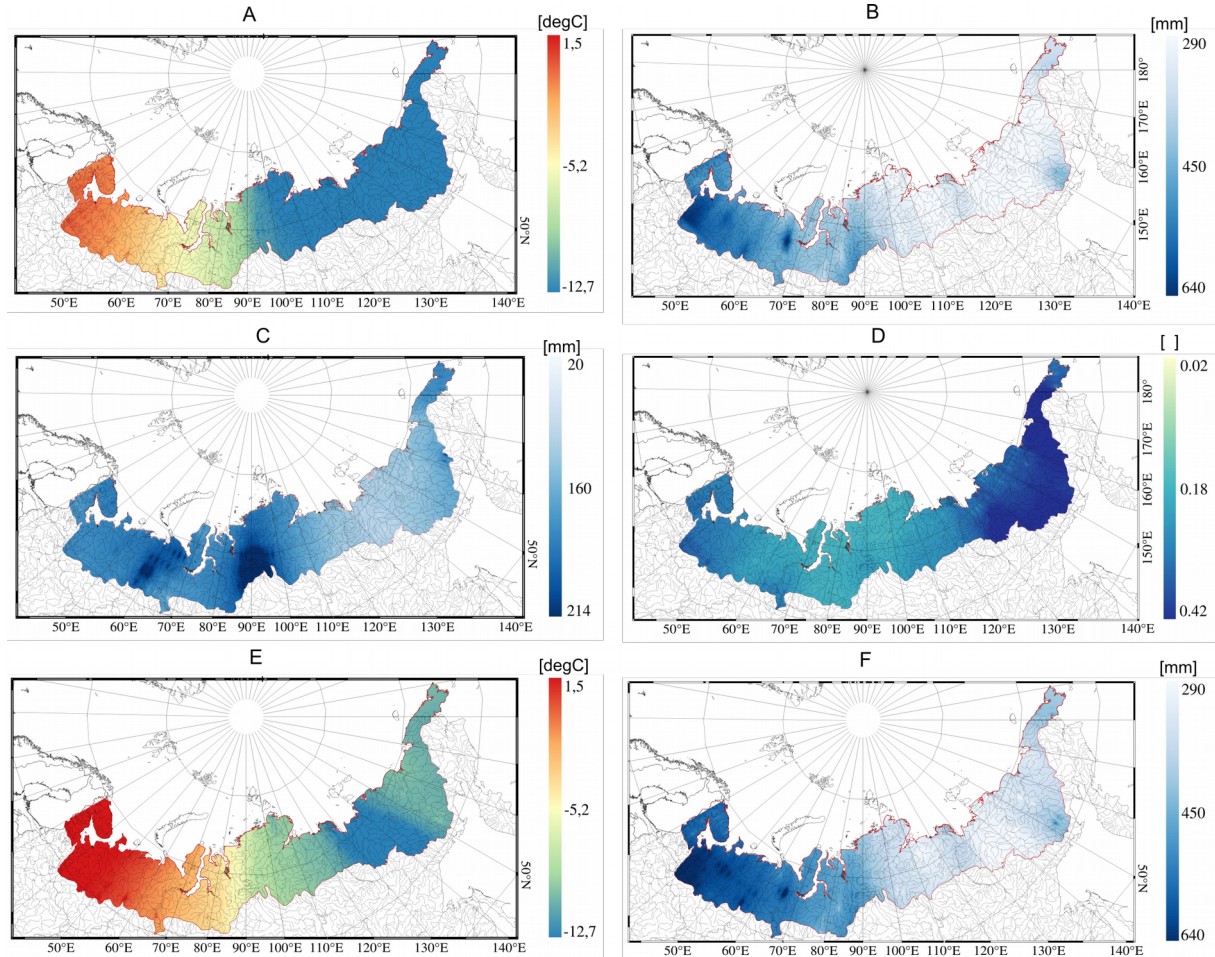

**Figure 4.** The datasets used in the study: A – the mean values of the annual air temperature for the reference period (Radionov and Fetterer, 2003, Catalogue of climatology, 1989); B – the mean values of the annual precipitation amount for the reference period (Radionov and Fetterer, 2003, Catalogue of climatology, 1989); C – the mean values of the spring flood flow depth of runoff for the reference period (Vodogretskiy, 1986); D – the coefficients of variation of the spring flood flow depth of runoff for the reference period (Rogdestvenskiy, 1986); E – the mean values of the annual ait temperature for the projected period (2010– 2039) under the RCP 4.5, average of four GCMs (Taylor et al., 2012); D – the mean values of the annual precipitation amount for the projected period (2010– 2039) under the RCP 4.5, average of four GCMs (Taylor et al., 2012). The territory of the Russian Arctic is outlined according to Nikanorov et al. (2007).

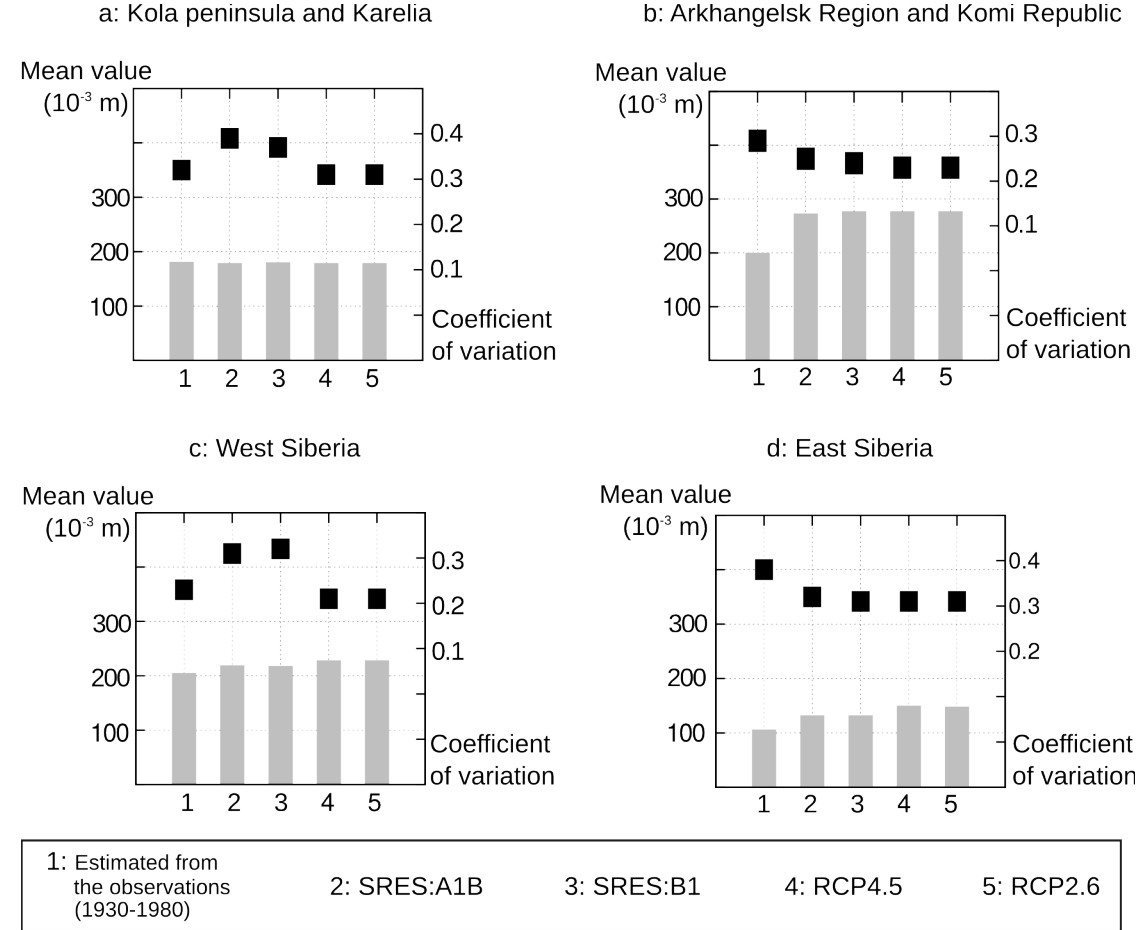

**Figure 5.** The changes of the mean values (bars) and coefficients of variation (squares) of the spring flood depth of runoff expected for the regions of the Russian Arctic for the period 2010–2039.

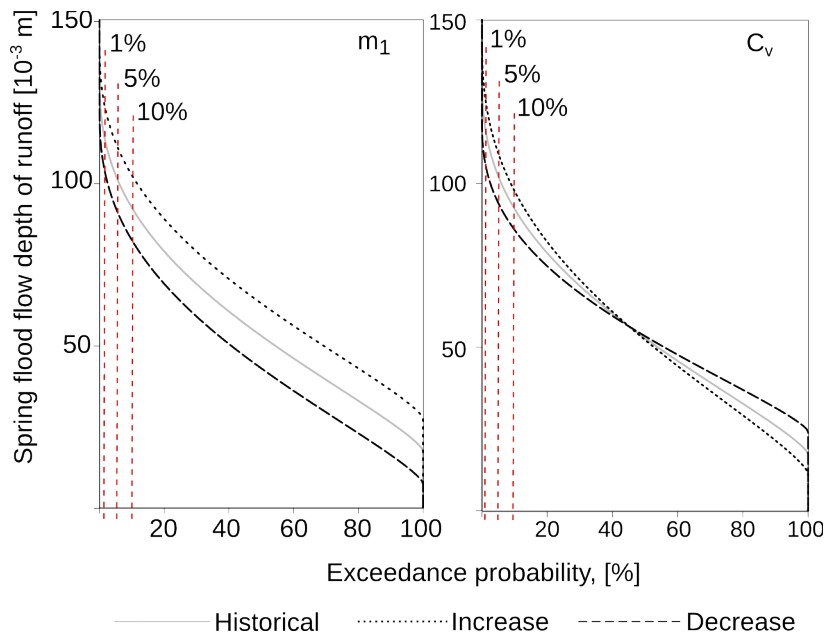

**Figure 6.** To illustrate the changes in the upper-tail values due to changes in the parameters of the PDF (mean values and coefficient of variation).

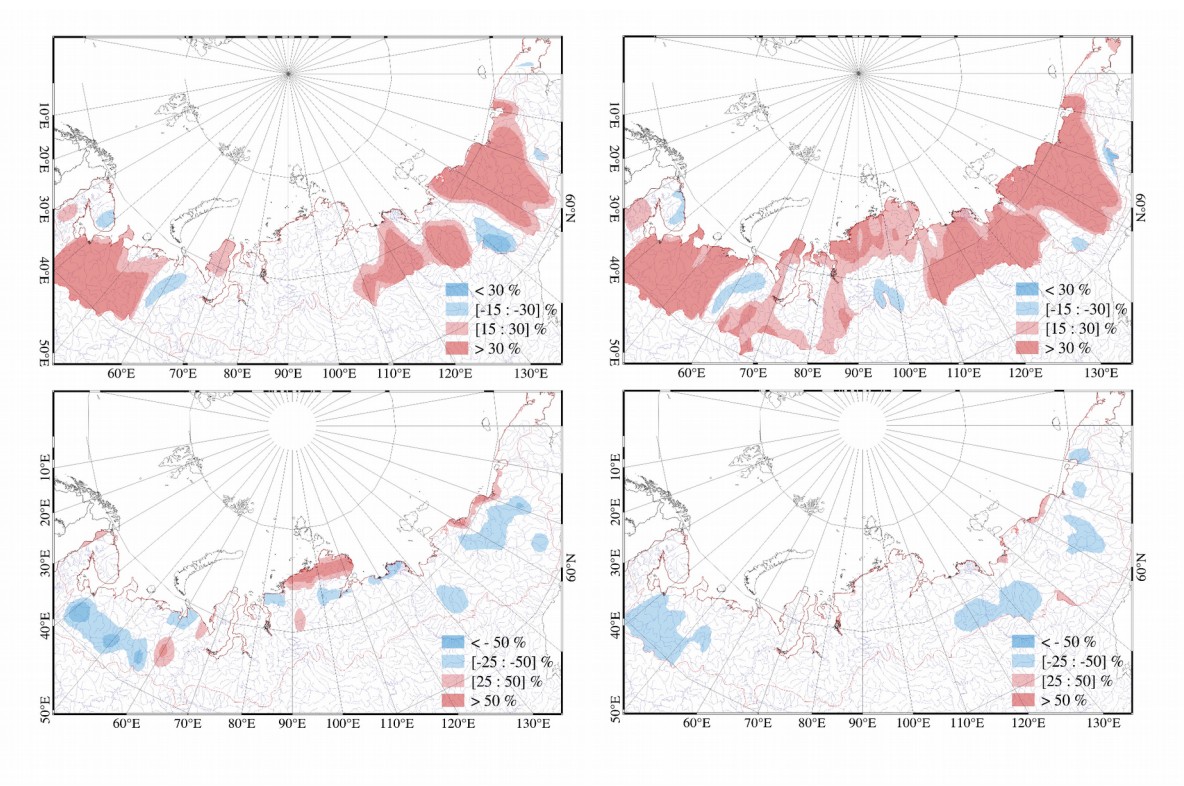

**Figure 7.** The regions with substantial changes in the mean values (top) and coefficients of variation (bottom) of the spring flood depth of runoff according to the MPIM:ECHAM5 under the SRES:B1 (left) scenario and the MPI-ESM-LR under the RCP 2.6(right).

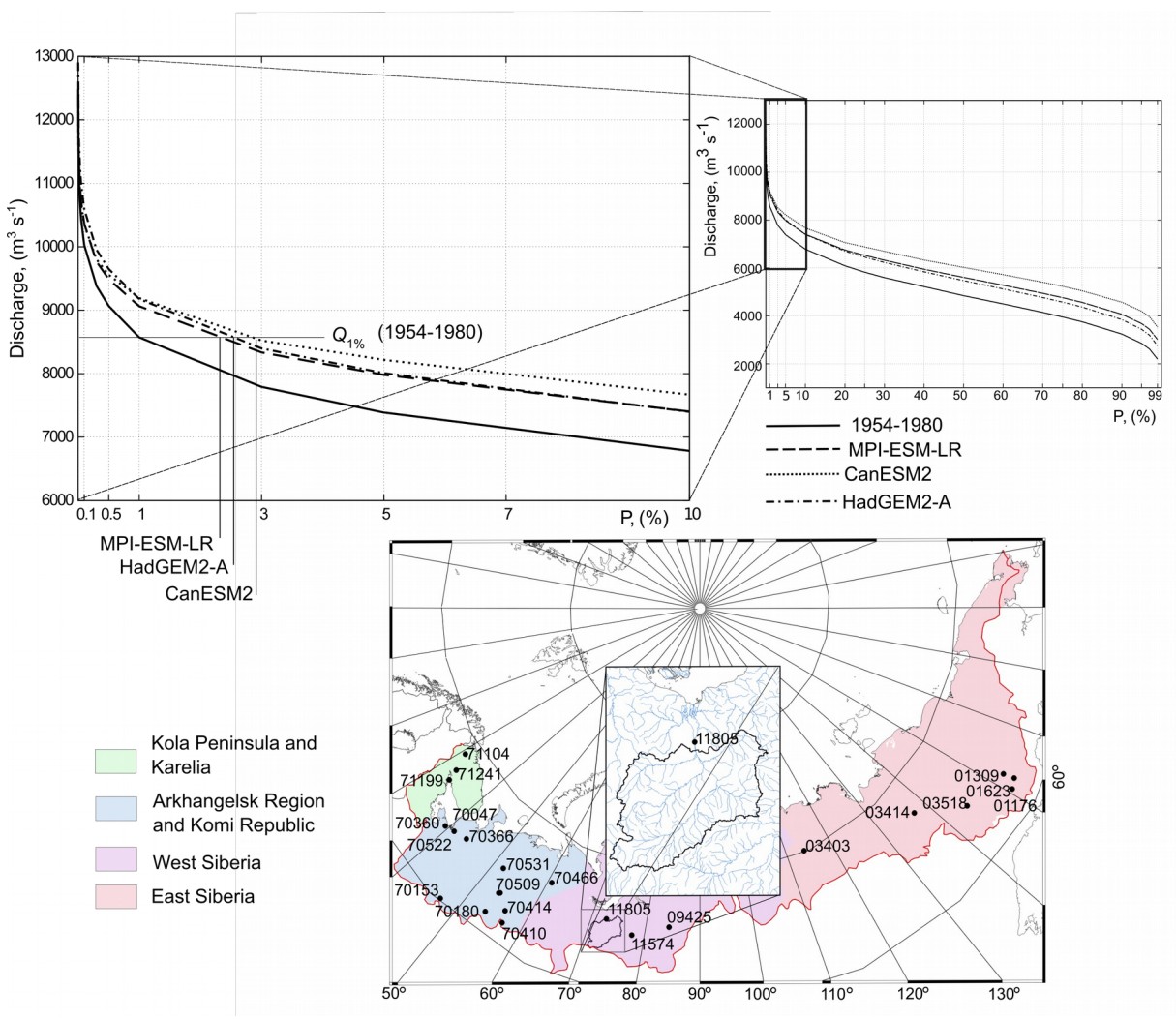

**Figure 8.** The exceedance probability curves of the peak-flow discharge for the period 1950–1980 and for the projected period 2010–2039 under the RCP 2.6 scenario (top) for the Nadym River at the Nadym City (11805): in the bottom figures the points and numbers correspond to the gauges used for the model (2) cross-validation.