# Peer review of "Assessment of extreme flood events in changing climate for a long-term planning of socio-economic infrastructure in the Russian Arctic"

_Hydrology and Earth System Sciences, 2015_

## Referee Comment (RC1) · Anonymous Referee #1 · 23 Feb 2016

General comments

This paper presents an approach to estimate spring flood depth of runoff in a future climate using a stochastic modeling framework. The main focus of the paper is an application of this method to evaluate changes for the Russian Arctic. The method builds on statistical properties of historical data in combination with estimates of expected changes that are derived from climate model simulations of several generations. The authors conclude that increases in spring flood depths but decreases in its variability are expected for much of the Russian Arctic. They also conclude that a main advantage of their approach is that it requires relatively little data and can be used in areas where detailed observations are not available.

[Figure]

The problem of estimating changing flood depths (or similarly, changing return periods of floods of a given magnitude) is certainly a challenge to engineering in many parts of the world. Stochastic approaches are quite common in engineering so the approach taken here could be possible to apply to infrastructure planning decisions. In general, the authors have used appropriate data for their climate projections, but in terms of the statistical and hydrological parameters, the data are drawn from Russian textbooks and reference material, which make them difficult to evaluate. The authors acknowledge this shortcoming, but I think it is a fundamental problem with the paper that also limits its applicability. Furthermore, it is not clear to me the degree of advance this method provides, in comparison with other methods.

Presently, the paper comes across as mainly a regional assessment of changing flood risks, which, although useful, I am not certain merits publication in HESS. The paper would be substantially improved (and more relevant to cite for other researchers) if the authors could show more clearly a) how their method compares with other stochastic approaches to evaluate change in flood risks, and b) how the method could be applied using other data sources than the reference tables available for the Russian basins they studied here.

Specific comments

P5 L1-17 The authors state that "[t]he stochastic approach was first proposed by Kovalenko (1993) and Kovalenko et al. (2010) simplified the basic stochastic model for applications of hydrological engineering". This is all the background we are given on stochastic approaches to river engineering under climate change. Surely there must be many other relevant contributions to research on stochastic hydrological modeling. The authors must here put their method in the context of the field, the state of the art, and how their method contributes to advancing the research frontier (if it does).

P5 L18-19 The authors state that "[t]he aim of this study is to perform a regional-scale assessment of the future extreme flood events based on climate projections for the

Russian Arctic". This is fine, but as mentioned above, I am not sure HESS is an ideal outlet for a study with such an aim.

P10 L8 It is not clear what the authors mean here. From the text it reads as if the model's prediction scores should be shown in Table 1, but I think the authors mean that the whole dataset is shown in Table 1. It must be clear why the authors refer the reader to look at Table 1 here.

P11 L17 It is not explained why there needed to be a statistically significant shift in the historical time series in order to consider it.

P11 L25 I am not too fond of personal communication when it comes to data references. Could these data not be obtained elsewhere? It means then that this study quite difficult for others to reproduce. Also, what reference is meant by "the multi-year catalogues of climatology (e.g. 1989)"?

P12 L28-29 It is not clear which statistical moments are meant here. The two moments shown in the table are explicitly not the same, so I am not sure what the authors mean by saying that they are assumed to be constant. Reading further, I assume this refers to a case or scenario where the authors used this assumption ("no model" case), but it is not clear from the text.

P12 L7-9 It is not correct to say that these are "5-10%" higher. Rather, the results are 5-10 percentage points higher.

P12 L20 Was there any reason for selecting these models? Previous research shows that the choice of climate model greatly influences the results of any hydrological model simulation that uses the GCM results. I strongly suggest the authors motivate why these models were selected. Are they representative of the whole ensemble, or is the sample perhaps biased in terms of key model aspects (climate sensitivity, hydrological response to temperature increase, etc)? To be clear, I do not think it necessary that the models be representative, but the reader should know why the models were selected

and if there are any aspects of these particular models that could influence the result. For instance, if the models all are in the low end of the climate sensitivity range, the results obtained in this study could be overly conservative (or vice versa of course in the case of very sensitive models). This is important information for the interpretation of the results, not to say any practical application of them.

P12 L25 What periods?

P13 L5 The procedure of extraction from maps should be described much clearer.

P14 L5 It needs to be made clear to what degree these results were already published. If the figures used here are directly from some previously published report, they should be part of the methods and data, and not a result of the study. If the authors analysed spatial datasets for new geographical regions, it is ok to have them as new results here. But it is not clear what the authors mean.

P14 Results from Table 4 It is difficult to follow the discussion of the results when they are presented in absolute values in the table, but discussed in terms of percentage increases in the text. I would prefer the authors either discussed also the absolute changes in the text, or that the percentage changes are shown in the table, so that one can follow which figures in the table are discussed in the text.

P14 L22-23 "The strongest increase (over 27 %) of the mean values with a lowest decrease of the coefficients of variation (over 17 %) is predicted by CaESM2 for the RCP2.6 scenario." I can't follow this from the table – which parameter increase of 27% are the authors talking about? Precipitation, temperature, or spring flood depth?

P14 L25-27 Here the authors talk about the "European part of the Arctic", and refer to figure 3, where there is one region referred to as "Northern European Arctica [sic]". There is also another region termed "Kola peninsula and Karelia". Do the authors by "European part of the Arctic" refer to only the "Northern European Arctica", or to both these regions? If so, it would be good to state this, for instance by labeling the panels

in Fig 3 as a, b, c, d, and then here refer to (Fig 3a-b) or similar. Not everyone knows where the Kola Peninsula and Karelia is. Furthermore, I assume this still only refers to territory within the Russian Federation, and it is therefore technically incorrect to refer to it as the "European part of the Arctic". Such a region would include parts of Scandinavia as well. I suppose that calling this the "European part of the Russian Arctic" would be more correct. In general, it would be helpful if the authors referred to the sub-regions they define in a consistent way throughout the paper, both in text and figures, and also clearly outlined these on a map.

P15 L16-18 Here it is a bit difficult to follow what Hirabayashi et al found. From the text I assume they found a decrease, but of what magnitude? How do the results really compare?

P16 L17 What Strategy? Please explain.

Table 3 It is not clear what the percentage refers to. All periods? All basins? How many value pairs are compared?

Figure 1 The figure should indicate the critical value of the t-test for the chosen significance level. Is this the dotted line in the figure? Please label this line.

Figure 5 must be improved. It is very difficult to see the patterns, and what pattern that corresponds to which value. I would suggest using grey shading instead for at least some of the categories, so that one does not have to use so many different patterns that are difficult to distinguish on the map.

Figure 6 must also be improved for the same reasons. Also, the ordering of the figures should be the same as the order they are referred to in the text.

References P23 "Government development strategy..." I was not able to retrieve this file from the web link provided.

Language and other minor points

[Figure]

Although the language is generally acceptable, there are quite a number of grammatical errors, and the paper needs editing before it can be accepted for publication. I have not noted all language points but list some issues that I noted here.

P3 L14 Remove "an" before "annual"

P3 L21 Replace "increases" with "increase"

P4 L8 Replace "this studies" with "these studies is"

P4 L9 Replace "calculation" with "calculation of"

P5 L29 Consider the choice of the word "alarm". I suggest "a warning", or more neutrally, "a highlight".

P9 L5 Replace "iterator" with "an iterator"

P10 L9 Replace "The example" with "An example"

P10 L10 Replace "partition of" with "partition"

P10 L13 Replace "1-year incrementing" with "increments of 1 year"

P10 L14 Replace "critical of 0.05 level of the statistical significance" with "t critical value at the 0.05 level of statistical significance"

P11 L13 Replace "average" with "an average"

P11 L14 Remove two occurrences of "the" on this line

P11 L15 Replace "66% cases" with "66% of cases"

P11 L16 Replace "contain the" with "have"

P11 L16 Replace "then" with "than"

P11 L19 Replace "corresponded" with "corresponding"

P11 L21 Replace "per a" with "per"

P11 L28 Replace "parameters" with "parameter"

P12 L1 Replace "the one" with "one"

P12 L6 and onward Please be consistent with the tense you are using. The discussion here should refer to the past tense and not the present.

P12 L9 Replace "gives the advantage of over 11–22 % in the percentage of the successful nominally predicted PDFs" with "gives an even larger advantage, with values 11–22 percentage points higher in terms of successful nominally predicted PDFs"

P12 L10 Replace "regional-oriented" with "the regional-oriented"

---

## Referee Comment (RC2) · F. Serinaldi (Referee) · 9 Mar 2016

**General comments**

The manuscript under review uses a simple linear system of two equations (Eq. 2 in the text), which is the simplified solution of an equivalent Fokker-Plank stochastic differential equation, to update the estimates of the first two moments of the flood depth distribution by replacing historical average of annual rainfall amount over a reference historical period ($\bar{N}_r$) with the corresponding average over a future time window ($\bar{N}_f$). Since Eq. 2 provides a relationship between rainfall and runoff, the idea is to use the rainfall output of climate models as an exogenous variable to update the parameters

of the flood depth distribution under climate (deterministic) change scenarios. I have to say that the paper is not very easy to read. Simple concepts are reported in a unnecessary complicated manner, while some parts are not necessary and only make the reading more difficult.

My doubts about such a type of papers are always the same: what about uncertainty? Why 50 years (data points) of annual maxima or spring flood data are not enough for standard flood frequency analysis but are enough when we deal with nonstationarity and projections, which in turn introduce further uncertainty? In this manuscript the Authors deal with time series lengths varying from 26 to 77 years with average of 51 years. For 50 observations, under iid conditions, the unbiased point estimate of the probability of exceedance of the largest observation is 1/50, but its 95% confidence interval is about (1/15, 1/2000), meaning that the return period ranges within 15 and 2000 years. Now under this condition of deep uncertainty, discussing changes of 7% in the point estimate of 1% Pearson III quantile, or changes "from 1% (calculated from the observations) to 2.5% (calculated according to the averaged climate projections)." for the exceedance probability of the largest observation is only a matter of speculation, whereby the Nadym River data consist of 37 annual values from 1955 to 1991. These changes are much lower than the large uncertainty affecting the point estimates based on historical records and easily fall within their confidence intervals. In this respect, all the moments, parameters and design quantiles reported throughout the text should be complemented by confidence intervals or something similar in order to communicate the actual uncertainty of the point estimates. The same holds for the CDFs shown in Figs. 2, 4, and 6, as well as mean values and coefficients of variation reported in Fig. 3. Simple standard bootstrap techniques can be used to accomplish this task.

By the way, since the Bullettin 17b is mentioned in the paper, I would like to highlight that it relies on regionalization procedures aiming at 'selling space for time' in order to (try to) reduce the large uncertainty of at-site estimates (as well as allowing for estimation in ungaged sites). This is just to say that at-site methods have been recognized

to provide unreliable extrapolations toward extreme (design) quantiles several decades ago under iid conditions; so, it seems to me quite anachronistic to propose at-site methods based on few tens of data to support even more uncertain 'nonstationary' or 'quasi-stationary' design procedures.

Other aspects rise further doubts. For example, the proposed system in Eq. 2 implies changes in both mean and variance, but the two-fold cross-validation is based on t-tests checking only for changes in mean. I cannot see information about possible (detected) changes in variance in the historical records. Moreover, from a modeling point of view, detected changes require attribution to avoid incorrect treatment of deterministic changes. Temperature is mentioned in the text but I cannot see where it is used in Eqs. 1 and 2 or elsewhere.

Finally, a minor remark about the introduction. This the n-th time I read this sentence in an introduction 'However, the frequency and magnitude of extreme flood events based on historical data do not provide correct estimations for a future under changing climate (Milly et al., 2008)'. Now, we can discuss for hours about the meaning of 'changing' in this context, but the main point is that this sentence is more or less copied and pasted as is from paper to paper. I would like to stress that Milly et al. (2008) is a one page opinion paper with no diagrams, proofs, equations, and only few references. It is perfectly legitimate but also debatable. I suggest thinking more critically before reporting sentences that are taken for granted as a truth (*aletheia*), when they are debatable opinions (*doxa*). In this respect, it is fairer to report different points of view, so that the readers can build their own. See e.g.:

Koutsoyiannis D, Montanari A, Negligent killing of scientific concepts: the stationarity case Hydrol Sci J (2014) http://dx.doi.org/10.1080/02626667.2014.959959,

Milly, P. C. D., J. Betancourt, M. Falkenmark, R. M. Hirsch, Z. W. Kundzewicz, D. P. Lettenmaier, R. J. Stouffer, M. D. Dettinger, and V. Krysanova (2015), On Critiques of "Stationarity is Dead: Whither Water Management?," Water Resour. Res., 51, 7785-

7789, doi:10.1002/2015WR017408.

Lins Harry F.,Timothy A. Cohn (2011) Stationarity: Wanted Dead or Alive? Journal of the American Water Resources Association 47(3), 475-480

Montanari A, Koutsoyiannis D, Modeling and mitigating natural hazards: Stationarity is immortal! Water Resour Res, 50 (12) (2014), pp. 9748-9756

Stedinger Jery R., Veronica W. Griffis (2011) Here to Where? Flood Frequency Analysis and Climate 47(3), 506-513

It might also be useful to recall Russell's principles ('On the value of scepticism' from 'The will to doubt'):

'The scepticism that I advocate amounts only to this:

- that when the experts are agreed, the opposite opinion cannot be held to be certain;

- that when they are not agreed, no opinion can be regarded as certain by a non-expert; and

- that when they all hold that no sufficient grounds for a positive opinion exist, the ordinary man would do well to suspend his judgment.'

To summarize, even though I understand the requirement of developing simple methods for practitioners, in my opinion, the method suggested in this study is too simple and overlooks the large uncertainty involved in this type of studies as well as several other aspects. I think that a better approach is to provide fair stationary analyses based on observations, quantify the uncertainty, understand if possible changes due to climate projections are significant (taking the projection uncertainty/reliability into account), and provide a set of possible values to be shrunk at a second stage via e.g. ex post economic/financial analyses.

Sincerely,

Francesco Serinaldi

---

## Author Comment (AC1) · 16 Mar 2016

We thank the referee for his/her questions, comments and suggestions. They were very useful in further improving of the manuscript. The Referee's comments are copied below, and our responses are written after each comment.

General comments This paper presents an approach to estimate spring flood depth of runoff in a future climate using a stochastic modeling framework. The main focus of the paper is an application of this method to evaluate changes for the Russian Arctic. The method builds on statistical properties of historical data in combination with estimates of expected changes that are derived from climate model simulations of several generations. The authors conclude that increases in spring flood depths but decreases in its

[Figure]

variability are expected for much of the Russian Arctic. They also conclude that a main advantage of their approach is that it requires relatively little data and can be used in areas where detailed observations are not available.

The problem of estimating changing flood depths (or similarly, changing return periods of floods of a given magnitude) is certainly a challenge to engineering in many parts of the world. Stochastic approaches are quite common in engineering so the approach taken here could be possible to apply to infrastructure planning decisions. In general, the authors have used appropriate data for their climate projections, but in terms of the statistical and hydrological parameters, the data are drawn from Russian textbooks and reference material, which make them difficult to evaluate. The authors acknowledge this shortcoming, but I think it is a fundamental problem with the paper that also limits its applicability.

Our response: The study presents long-term assessment of extreme flood events only for the Russian part of the Arctic. Thus, the runoff data were extracted from the official issues of State Water Cadastre by Roshydromet. These data passed the quality control allowing to use the runoff time series in calculation of extreme flood events for engineering purposes. The runoff data are available by the order to the State Hydrological Institute (www.hydrology.ru). The method presented in the paper can be applied in other regions as well. Then, to calculate multi-year time series of spring flood flow depth (or peak flow), the daily discharge time series are required. Global and regional runoff databases may be also used to evaluate the regional and basin scale assessment of the future floods with required probability of exceedance using the climate change projections. Examples of such datasets include the following: 1. The Global Runoff Data Centre, Koblenz, Germany 2. The Environmental Information System (HERTTA) database at the Finnish Environment Institute. 3. Vattenwebb dataset at the Swedish Meteorological and Hydrological Institute. We have improved the manuscript accordingly.

Furthermore, it is not clear to me the degree of advance this method provides, in com-

parison with other methods.

Our response: Physically-based hydrological models (even with stochastic components) generate the flow time series (or runoff signal) based on the time series of meteorological variables. Thus, to estimate the extreme hydrological events (floods or droughts) with required probability of exceedance for a single catchment, one should run the physically-based catchment-scale hydrological model for a particular climate scenario (or a set of scenarios) and simulate the runoff signal. In case of the hydrological model with stochastic components, the signal could be performed by Monte-Carlo simulations within a-priory defined random generators. In performing of regional scale flood (or drought) frequency analysis using climate projections, the runoff signal should be simulated for a set of watersheds. It makes the calculations computationally expensive, especially in case of the climate ensembles. The approach presented in this paper allows to skip the simulations of the runoff signal since only the parameters of pdf are directly simulated from the meteorological mean values (for projected periods of 20-30 years). These parameters are further used to model the pdf within a theoretical distribution, allowing to evaluate the flood events with required probability of exceedance in the future. Thus, it makes it easy to perform regional-scale assessments of the detrimental hydrological events (floods and droughts) in the future, presented by a particular scenario or by ensembles of the climate projections. We have corresponding additions and improvements in the revised manuscript.

Presently, the paper comes across as mainly a regional assessment of changing flood risks, which, although useful, I am not certain merits publication in HESS. The paper would be substantially improved (and more relevant to cite for other researchers) if the authors could show more clearly a) how their method compares with other stochastic approaches to evaluate change in flood risks, and b) how the method could be applied using other data sources than the reference tables available for the Russian basins they studied here.

Our response: (a) The basic profits of the method applied are: 1. a low number of forcing and simulated variables (only statistical moments of climate and hydrological variables are needed); 2. a low number of parameters (physical processes described integrally by a lumped hydrological model); 3. a relative simplicity of a regionally-oriented parameterization. Thus, the stochastic model used is extremely cheap computationally and allows to provide regional-scale assessment of extreme floods events in the future. Then, the warning regions where the risks to damage the social infrastructure increase may be outlined. We added this information in the revised manuscript.

(b) For other regions, the steps of the modelling are following: 1. The multi-year time series of a yearly maximum runoff (discharges or spring flood flow depth of runoff) are calculated from the daily runoff time series. 2. The mean values, the coefficients of variation and skewness are estimated from the observed time series of a yearly maximum runoff. Also, the mean values are estimated for the annual precipitation amount and air temperature for the selected period (considered as reference). 3. The numerical values of the model parameters are evaluated from Eq. (3). 4. The mean values of the annual precipitation amount and air temperature for the future period are evaluated from the climate projections. 5. The future mean values, the coefficients of variation and skewness of an yearly maximum runoff are evaluated using the mean values of the annual precipitation amount and air temperature with Eq. (4). To perform the model cross-validation and to develop the regional-oriented parameterization scheme, the multi-year time series of a yearly maximum runoff with the periods of statistically significant shifts in the mean values and coefficient of variations are required. We added this information in the revised manuscript.

Specific comments

P5 L1-17 The authors state that "[t]he stochastic approach was first proposed by Kovalenko (1993) and Kovalenko et al. (2010) simplified the basic stochastic model for applications of hydrological engineering". This is all the background we are given on stochastic approaches to river engineering under climate change. Surely there must be many other relevant contributions to research on stochastic hydrological modeling.

The authors must here put their method in the context of the field, the state of the art, and how their method contributes to advancing the research frontier (if it does).

Our response: The description of the general context of the stochastic approach used in the study was expanded, and the corresponding references are provided. We added the details in the revised text of the manuscript, the reference list was expanded with: 1. Kritsky, S.N. and Menkel, M.F.: 1946. On the methods of studying the random variations of river flow, Gidrometeoizdat, Leningrad. Kite, G.W. 1977: Frequency and risk analysis in hydrology. Water Resour. Publications. Colorado: Fort Collins, 224 pp. 3. Benson, M.A. 1968: Uniform flood frequency estimating methods for federal agencies. Water Resour. Res. 4, 891–908. 4. Elderton, Sir W.P, Johnson, N.L. 1969: Systems of Frequency Curves. Cambridge University Press, London, 224 pp.

P5 L18-19 The authors state that "[t]he aim of this study is to perform a regional-scale assessment of the future extreme flood events based on climate projections for the Russian Arctic". This is fine, but as mentioned above, I am not sure HESS is an ideal outlet for a study with such an aim.

Our response: The study aims to perform the spatial and temporal characteristics of the regional water resources with particular focus on the Russian part of Arctic. The region was chosen since its sustainable development in changing climate is important for the Russian Federation from the economical point of view (and the study was supported by the Ministry of Education and Science). However, the stochastic approach applied can be used also to perform global as well as catchment (see example in fig. 6) scales assessment of flood events with required probability of exceedance. The paper presents the simplified model, the method of its validation and the regional oriented parameterization scheme (Arctic). The general statements of the stochastic approach used have been already presented in the previous papers (unfortunately mostly in Russian, but there are at least two papers in English by Kovalenko, 2014 and Domínguez and Rivera, 2010). From our point of view, the paper fits to scopes 2 and 3 of the HESS journal, as defined at http://www.hydrology-and-earth-system-sciences.net/about/

P10 L8 It is not clear what the authors mean here. From the text it reads as if the model's prediction scores should be shown in Table 1, but I think the authors mean that the whole dataset is shown in Table 1. It must be clear why the authors refer the reader to look at Table 1 here.

Our response: Yes, the reference to Table 1 was removed in the revised version.

P11 L17 It is not explained why there needed to be a statistically significant shift in the historical time series in order to consider it.

Our response: The parameters of pdf of multi-year runoff the future time periods are simulated by the model using the parameters of pdf evaluated for the present period (the initial conditions of the model). The hypothesis of quasi-stationarity considers the present and future time periods as two steps with different (statistically significant) values of the pdf's parameters. Thus, to verify the model using historical data, two periods with different values of the pdf's parameters are required to perform the cross-validation procedure. For the Pearson III type distributions the pdf's parameters can be calculated using the mean value, coefficient of variation and skewness. Therefore, we analyzed observed time series to define two periods with statistically significant difference at least for the mean value (t-test). These time series were used to verify the model. We have corresponding additions and improvements in the revised manuscript.

P11 L25 I am not too fond of personal communication when it comes to data references. Could these data not be obtained elsewhere? It means then that this study quite difficult for others to reproduce. Also, what reference is meant by "the multi-year catalogues of climatology (e.g. 1989)"?

Our response: Yes, we agree that it is not a good idea to furnish the personal communication as the reference to the data source. Thus, the reference to the Meteorological Data from the Russian Arctic (1961-2000, Version 1. 2003. Edited by V. F. Radionov and F. Fetterer. National Snow and Ice Data Center. Boulder, Colorado USA. NSIDC: National Snow and Ice Data Center. http://dx.doi.org/10.7265/N56H4FB3) was added,

which was originally created based on the dataset of the Arctic and Antarctic Research Institute (St. Petersburg). The personal communication (N. Bryazgin) just expands the collection in space and time. The multi-year catalogs of climatology are official editions by Roshydromet, and these data are usually used in engineering calculations. There are several issues covering the data from different regions. However, using these data sources does not restrict the application of the method, and the global dataset of the climatological data can be used (i.e. CRU dataset from http://www.cru.uea.ac.uk/data). We have corresponding additions and improvements in the revised manuscript.

P12 L28-29 It is not clear which statistical moments are meant here. The two moments shown in the table are explicitly not the same, so I am not sure what the authors mean by saying that they are assumed to be constant. Reading further, I assume this refers to a case or scenario where the authors used this assumption ("no model" case), but it is not clear from the text.

Our response: The term "statistical moments ..." used in the text because of the Eq. (2) is written for the initial statistical moments. These moments are further used to calculate the mean values and coefficients of variation. "No model" represents the case, when the climate change is not taken into account, and thus the parameters of pdf are not modified for the period of prediction. This case shows the situation considering in the guidelines for the engineering hydrology (Bulletin 17-B), which used only observed time series to evaluate the parameters of pdf. We have corresponding additions and improvements in the revised manuscript.

P12 L7-9 It is not correct to say that these are "5-10%" higher. Rather, the results are 5-10 percentage points higher. Our response: Yes, the text was corrected.

P12 L20 Was there any reason for selecting these models? Previous research shows that the choice of climate model greatly influences the results of any hydrological model simulation that uses the GCM results. I strongly suggest the authors motivate why these models were selected. Are they representative of the whole ensemble, or is the

sample perhaps biased in terms of key model aspects (climate sensitivity, hydrological response to temperature increase, etc)? To be clear, I do not think it necessary that the models be representative, but the reader should know why the models were selected and if there are any aspects of these particular models that could influence the result. For instance, if the models all are in the low end of the climate sensitivity range, the results obtained in this study could be overly conservative (or vice versa of course in the case of very sensitive models). This is important information for the interpretation of the results, not to say any practical application of them.

Our response: The long-term assessment of extreme flood events was performed for each scenario and model from the Fourth and Fifth IPCC Assessment Reports and published as the final reports for the Ministry of Education and Science of the Russian Federation (2013-2015). This paper presents the results of flood frequency analysis obtained for the models that produced typical scenarios and models that were close to the regionally averaged scenario recommended by Gaidukova (2012). In this paper, the author provides the estimates of ensemble average scenario for the projections from the Fourth Assessment Report of IPCC. The GCMs used represent the climate projection close to the typical, and show that the hydrological modelling results do not vary much under the the climate forcing with the small differences. We have corresponding additions and improvements in the revised manuscript.

P12 L25 What periods?

Our response: The reference period of climatology was considered as 1961-90 for climatology. We added the details in the revised text of the manuscript.

P13 L5 The procedure of extraction from maps should be described much clearer.

Our response: The technical details of this procedure were not included to the text of this paper, since the procedure is commonly used in GIS-applications without specific references except of user manuals for the particular software. Thus, the following description was added: "The procedure to obtain the mean values and coefficients of

variation from the maps included scanning of paper maps, georeferencing of images, data digitizing, and interpolation into the grid nodes of the particular GCM."

P14 L5 It needs to be made clear to what degree these results were already published. If the figures used here are directly from some previously published report, they should be part of the methods and data, and not a result of the study. If the authors analysed spatial datasets for new geographical regions, it is ok to have them as new results here. But it is not clear what the authors mean.

Our response: Yes, it is important to separate the previous results from the new ones. The studies by Govorkova and Meleshko include the climate changes assessment for the territories of the Russian Federation as a whole, and do not provide the estimates within the geographical domain of the Russian Arctic, which was outlined in this study according to the hydrological principles as suggested by Ivanov and Yankina (1992). Then, the results presented further in the text of manuscript have not been published in previous papers (except the technical reports). We have corresponding additions and improvements in the revised manuscript.

P14 Results from Table 4 It is difficult to follow the discussion of the results when they are presented in absolute values in the table, but discussed in terms of percentage increases in the text. I would prefer the authors either discussed also the absolute changes in the text, or that the percentage changes are shown in the table, so that one can follow which figures in the table are discussed in the text.

Our response: We added the absolute values of the changes into the text of the discussion.

P14 L22-23 "The strongest increase (over 27 %) of the mean values with a lowest decrease of the coefficients of variation (over 17 %) is predicted by CaESM2 for the RCP2.6 scenario." I can't follow this from the table – which parameter increase of 27% are the authors talking about? Precipitation, temperature, or spring flood depth? Our response: This is for the spring flood flow depths. We corrected the text to provide

clear descriptions in this part of the manuscript.

P14 L25-27 Here the authors talk about the "European part of the Arctic", and refer to figure 3, where there is one region referred to as "Northern European Arctica [sic]". There is also another region termed "Kola peninsula and Karelia". Do the authors by "European part of the Arctic" refer to only the "Northern European Arctica", or to both these regions? If so, it would be good to state this, for instance by labeling the panels in Fig 3 as a, b, c, d, and then here refer to (Fig 3a-b) or similar. Not everyone knows where the Kola Peninsula and Karelia is. Furthermore, I assume this still only refers to territory within the Russian Federation, and it is therefore technically incorrect to refer to it as the "European part of the Arctic". Such a region would include parts of Scandinavia as well. I suppose that calling this the "European part of the Russian Arctic" would be more correct. In general, it would be helpful if the authors referred to the sub-regions they define in a consistent way throughout the paper, both in text and figures, and also clearly outlined these on a map.

Our response: In corrected version we refer the panels of fig. 3 as a,b,c and d and correct the text in the discussion. Moreover, the fig. 6 presents the location of geographic domains, which were discussed in the fig. 3 and text.

P15 L16-18 Here it is a bit difficult to follow what Hirabayashi et al found. From the text I assume they found a decrease, but of what magnitude? How do the results really compare?

Our response: It is very difficult to compare our result with other studies because different flood characteristics are addressed. Only indirect and quantitative comparison is possible. For the comparison we assume that for Pearson Type III distributions, an increase of the mean values and the coefficients of variation leads to an increase of upper-tail values. Then, present 100-year floods occur more frequently (Fig. 4). Also, a decrease of the mean values and the coefficients of variation leads to a decrease of upper-tail values. In this case, we can expect 100-year floods decreased. For the

eastern part of the Arctic, an increase of historical 100-year maximum discharges is predicted by Hirabayashi et al. (2008; 2013) under the SRES:A1B scenario for the period 2001–2030. This is in accordance with our results, we also expect an increase of upper-tail runoff values since the mean values and coefficients of variation were estimated to enlarge in average for this region. For the north-east European Arctic we expect a significant increase the frequency of present 100-year flood events. This is in contrast to Hirabayashi et al. (2013), which presents the global scale estimates of the projected change in flood frequency. The flood frequency is decrease in many regions of northern and eastern Europe according to Hirabayashi et al. (2013). The feasible reason of such disagreement is the spatial coarseness of the model used by Hirabayashi et al. (2013). The model is calibrated using the observations from the watersheds larger than 100,000 km2. We added the details to discuss the comparison of our results with previously obtained in the revised text of the manuscript.

P16 L17 What Strategy? Please explain.

Our response: The Strategy is the official document of the Government of the Russian Federation, it is more political (not research) issue. We exclude this reference in the revised version of the paper.

Table 3 It is not clear what the percentage refers to. All periods? All basins? How many value pairs are compared?

Our response: We provide the details about the percentage of the successful "nominal prediction" (used to perform the model cross-validation) in the text and in the header of the table 3.

Figure 1 The figure should indicate the critical value of the t-test for the chosen significance level. Is this the dotted line in the figure? Please label this line. Our response: The explanation was added on the text under Figure1

Figure 5 must be improved. It is very difficult to see the patterns, and what pattern that

corresponds to which value. I would suggest using grey shading instead for at least some of the categories, so that one does not have to use so many different patterns that are difficult to distinguish on the map. Figure 6 must also be improved for the same reasons. Also, the ordering of the figures should be the same as the order they are referred to in the text.

Our response: Figures 5 and 6 were improved with the regions presented by color patterns.

References P23 "Government development strategy. . ." I was not able to retrieve this file from the web link provided.

Our response: The reference to the Strategy was removed.

Language and other minor points Although the language is generally acceptable, there are quite a number of grammatical errors, and the paper needs editing before it can be accepted for publication. I have not noted all language points but list some issues that I noted here.

Our response: We corrected the revised text to exclude the grammatical mistakes mentioned in the list.

---

## Author Comment (AC2) · 16 Mar 2016

We thank the referee, Dr. Francesco Serinaldi, for his constructive comments and suggestions. They were very useful in further improving of the manuscript.

The Referee's comments are copied below, and our responses are written after each comment.

General comments The manuscript under review uses a simple linear system of two equations (Eq. 2 in the text), which is the simplified solution of an equivalent Fokker-Plank stochastic differential equation, to update the estimates of the first two moments of the flood depth distribution by replacing historical average of annual rainfall amount

over a reference historical period () with the corresponding average over a future time window (). Since Eq. 2 provides a relationship between rainfall and runoff, the idea is to use the rainfall output of climate models as an exogenous variable to update the parameters of the flood depth distribution under climate (deterministic) change scenarios. I have to say that the paper is not very easy to read. Simple concepts are reported in a unnecessary complicated manner, while some parts are not necessary and only make the reading more difficult.

My doubts about such a type of papers are always the same: what about uncertainty? Why 50 years (data points) of annual maxima or spring flood data are not enough for standard flood frequency analysis but are enough when we deal with nonstationarity and projections, which in turn introduce further uncertainty? In this manuscript the Authors deal with time series lengths varying from 26 to 77 years with average of 51 years. For 50 observations, under iid conditions, the unbiased point estimate of the probability of exceedance of the largest observation is 1/50, but its 95% confidence interval is about (1/15, 1/2000), meaning that the return period ranges within 15 and 2000 years. Now under this condition of deep uncertainty, discussing changes of 7% in the point estimate of 1% Pearson III quantile, or changes "from 1% (calculated from the observations) to 2.5% (calculated according to the averaged climate projections)." For the exceedance probability of the largest observation is only a matter of speculation, whereby the Nadym River data consist of 37 annual values from 1955 to 1991. These changes are much lower than the large uncertainty affecting the point estimates based on historical records and easily fall within their confidence intervals. In this respect, all the moments, parameters and design quantiles reported throughout the text should be complemented by confidence intervals or something similar in order to communicate the actual uncertainty of the point estimates. The same holds for the CDFs shown in Figs. 2, 4, and 6, as well as mean values and coefficients of variation reported in Fig. 3. Simple standard bootstrap techniques can be used to accomplish this task.

Our response: To calculate the confidence intervals not only including size of sample

and level of confidence are important. Also a population variability is required. How the numbers shown in your sentence were evaluated (1/15, 1/2000)? Of course, a larger sample size normally will lead a better estimate of the population parameters. However, engineering hydrology usually operates with samples with lengths of the order of 50–80 years, and with a statistically significant autocorrelation detection in the observed time series. The pdfs of multi-year runoff (annual, maximal and minimal) have significant skewness and do not fit Pearson I type distribution. Strictly speaking, the runoff is not i.i.d. variable. However, numerous studies provide the practical aspects of application of the classical statistical methods to estimate the risk of occurrence of detrimental hydrological events, and flood frequency analysis in particular. We have added references to the revised manuscript, to put the study in general context as recommended by the Anonymous Reviewer #1. These studies also include discussion of the confidence intervals for the runoff calculations. We thank the Referee for so sharply formulating the main idea of the method: to predict the future parameters of pdf using new climatology, and to construct the pdf with a-priory defined distribution (Pearson III type), and finally to calculate the tailed values. However, in this case the confidence intervals can not be used to present the uncertainties (the credible intervals may be more suitable).

By the way, since the Bullettin 17b is mentioned in the paper, I would like to highlight that it relies on regionalization procedures aiming at 'selling space for time' in order to (try to) reduce the large uncertainty of at-site estimates (as well as allowing for estimation in ungaged sites). This is just to say that at-site methods have been recognized to provide unreliable extrapolations toward extreme (design) quantiles several decades ago under iid conditions; so, it seems to me quite anachronistic to propose at-site methods based on few tens of data to support even more uncertain 'nonstationary' or 'quasi-stationary' design procedures.

Our response: It was mentioned in the introduction that the study presents the method to evaluate a regional scale assessment of extreme flood events. The Bulletin-17B and

the Russian Guidelines (SP33-101-2003) are also dedicated to regionalization using the observed time series (and sampled statistics). The present study used the same concept, but for the projected parameters of pdf, since future time series do not exist. The regionalization concept may seems anachronistic, but this circumstance allows easy performing a regional scale evaluation of extreme detrimental events for the risk assessment purposes.

Other aspects rise further doubts. For example, the proposed system in Eq. 2 implies changes in both mean and variance, but the two-fold cross-validation is based on t-tests checking only for changes in mean. I cannot see information about possible (detected) changes in variance in the historical records. Moreover, from a modeling point of view, detected changes require attribution to avoid incorrect treatment of deterministic changes.

Our response: It is not easy to find the time series with two periods with statistically significant differences in two moments (in Eq.2). The main problem was already mentioned by the Referee: the length of time series does not allow to perform the sub-division based of variance analysis (F-test) even used the statistical estimators "adapted" to engineering calculations. Then, we used the assumption, that for the runoff time series, the difference in the second statistical moment is proved by the statistical significance of the differences in the first statistical moment.

Temperature is mentioned in the text but I cannot see where it is used in Eqs. 1 and 2 or elsewhere.

Our response: The mean values of air temperature were used to perform the regional-oriented parameterization scheme of the model. This part of the study is presented in details within other paper (Shevnina, 2012), and was published unfortunately only in Russian (as numerous papers described the basic assumptions used by the method (Kovalenko, 1993, Kovalenko et al. 2006, 2010)). This circumstance makes difficulties during reading of the paper (as was mentioned by the Referee in his general comments) since the study tries to reconcile statistic and physical principles both used in this study.

Finally, a minor remark about the introduction. This the n-th time I read this sentence in an introduction 'However, the frequency and magnitude of extreme flood events based on historical data do not provide correct estimations for a future under changing climate (Milly et al., 2008)'. Now, we can discuss for hours about the meaning of 'changing' in this context, but the main point is that this sentence is more or less copied and pasted as is from paper to paper. I would like to stress that Milly et al. (2008) is a one page opinion paper with no diagrams, proofs, equations, and only few references. It is perfectly legitimate but also debatable. I suggest thinking more critically before reporting sentences that are taken for granted as a truth (aletheia), when they are debatable opinions (doxa). In this respect, it is fairer to report different points of view, so that the readers can build their own. See e.g.: Koutsoyiannis D, Montanari A, Negligent killing of scientific concepts: the stationarity case Hydrol Sci J (2014) http://dx.doi.org/10.1080/02626667.2014.959959, Milly, P. C. D., J. Betancourt, M. Falkenmark, R. M. Hirsch, Z. W. Kundzewicz, D. P. Lettenmaier, R. J. Stouffer, M. D. Dettinger, and V. Krysanova (2015), On Critiques of "Stationarity is Dead: Whither Water Management?," Water Resour. Res., 51, 7785- 7789, doi:10.1002/2015WR017408. Lins Harry F.,Timothy A. Cohn (2011) Stationarity: Wanted Dead or Alive? Journal of the American Water Resources Association 47(3), 475-480 Montanari A, Koutsoyiannis D, Modeling and mitigating natural hazards: Stationarity is immortal! Water Resour Res, 50 (12) (2014), pp. 9748-9756 Stedinger Jery R., Veronica W. Griffis (2011) Here to Where? Flood Frequency Analysis and Climate 47(3), 506-513

Our response: Yes, we agree that it is necessary to emphasize the doxa-status of the hypothesis of the non-stationarity in the context of hydrological applications using observed datasets. There are numerous studies which proof and deny both stationary and non-stationary hypotheses (we added discussion in the introduction of the revised manuscript). We consider it improbable that changes in meteorological variables would

remain unnoticed in runoff, which is an element of general water balance. From a practical point of view, the method allowing to evaluate the regional scale assessment of detrimental hydrological events is required besides of discussions of reality of the changes in climate, that why the studies similar to presented are usually supported in national and international levels.

It might also be useful to recall Russell's principles ('On the value of scepticism' from 'The will to doubt'): 'The scepticism that I advocate amounts only to this:  c that when the experts are agreed, the opposite opinion cannot be held to be certain;  c that when they are not agreed, no opinion can be regarded as certain by a nonexpert; and  c that when they all hold that no sufficient grounds for a positive opinion exist, the ordinary man would do well to suspend his judgment.'

Our response: Yes, skepticism is very useful, especially in science. But even in science, we need something (hypothesis, axioms, etc.) to rely on, to have a starting point to move forward. Otherwise, we will stay in a certain point, having no courage to move forward.

To summarize, even though I understand the requirement of developing simple methods for practitioners, in my opinion, the method suggested in this study is too simple and overlooks the large uncertainty involved in this type of studies as well as several other aspects. I think that a better approach is to provide fair stationary analyses based on observations, quantify the uncertainty, understand if possible changes due to climate projections are significant (taking the projection uncertainty/reliability into account), and provide a set of possible values to be shrunk at a second stage via e.g. ex post economic/financial analyses.

Our response: Yes, a fair stationary analysis based on observations is very important but not sufficient alone, since it does not provide a forecast, which is important for practical applications. In this study we try to fill the gap between engineering hydrology (with statistical methods) and physical hydrology (water balance) to provide forecasts of extreme flood events using climate projections. Again we note that the way of inter-comparison of the method used in this study with physically-based hydrological modelling is of a high interest.

---

## Author Comment (AC3) · 14 May 2016

**Response to Referee #3**

We thank the referee, for the constructive comments and suggestions. They were very useful in further improving of the manuscript. The Referee's comments are copied below, and our responses are written after each comment.

The article gives a huge amount of new data about the Arctic runoff and climate, can help to understand some conformity to natural processes and statistical regularities. The long data grids are used for the work. It allows to support the article for the publication. But the article should have a major revision. Restructuring of some chapters are also necessary.

**Main remarks for the paper**
1. The period of modeling in the paper is 2010-2039. 2016th is nowadays. Why the model has not been checked for 2010-2015 (2014)? It can be helpful for estimation the method adequacy.
Our response: The model simulates the multi-year statistical values (the mean and coefficient of variation). The 5-year period is completely insufficient to estimate these values to verify the model (2).

2. If the mean values increase (17-23%) and the Cv decrease (5-16 %) simultaneously it can mean the runoff magnitude uprising. Thus, it should be analyses in the article (in discussion chapter, for example).
Our response: The changes in the runoff multi-year statistics lead the alterations in the tailed values of the PDF: escalating (usually due to the increase in the mean and/or Cv), diminishing (usually due to the decrease in the mean and/or Cv) or neglecting (usually in case if the increase/decrease in the mean values is accompanied by the decrease/increase in Cv). The evaluation of the thresholds for the simultaneous changes in the mean and Cv, which lead these three types of the alteration in the tailed values of the PDF Pearson Type III is the issue have to be considered in the special statistical (not hydrological) study. In our study we rely on the simple assumption (P. 11 L. 10–15).

3. The Nadym River is one of the Russian Arctic Rivers, and not the lager one. There are no socio-economic aspects in the article results that had been analyzed according to the examined modeling evaluation. In the second part of the paper the Yana River is considering. These incompatibilities could be solved by changing of the paper title or addition river examples examining from the other Russian Arctic regions. It is actual in the case of examination of the aim of the paper study (Page 3, Line 23) – "…to perform a regional-scale assessment…"
Our response: The paper discusses the issues of the regional scale assessment the expected changes in the extreme floods for the Russian Arctic. The main result of the study is shown in Fig. 7, where the warning regions are outlined. However, the paper also provides two examples for the particular catchments. The first example illustrates the model cross-validation procedure (the Yana River), and the second example shows the practical application of the maps with warning regions to calculate the maximal discharges of low probability of exceedance for the Nadym River. The economic issues itself were not include into this study, which was mostly dedicated to the hydrological aspect.

4. The methods \ materials chapter has to be restructured and clarified.
Our response: This chapter was improved in the revised version of the manuscript according to the suggestions provided by all Reviewers. In particular, we (i) discussed the opposing views on the significance of affecting the climate change to the hydrological regime (P. 2, L. 3–13); (ii) provide the background, why there needed to be a statistically significant shift in the observed time series to perform the model cross-validation (P. 6, L. 22–29); (iii) discussed the dataset used in the study and references connected with data (P. 8, L. 10–13, P. 13, L. 3–8); (iv) explained the case of "no model" represents in the cross-validation section (P. 8 L. 28–31); (v) added description of the data sets used in the study (Fig. 4, P. 9, L. 5–13).

5. The link to the table 6 is earlier than to the tables 3-5 in the text. The table numeration has to be done sequentially.
Our response: We corrected the numeration of the tables and figures to be sequentially.

6. Some results of the paper are presented in the methods chapter and in the Introduction but not in the result chapter. There are some results without discussion or without explanation of methodic that leads to carried out it. Thus, the Figure 5 shows regions of the spring flood depth of runoff according two models calculation without comparison or other explanations.
Our response: We have revised the manuscript in conforming of the name of the sections and their content and improved the text. Also, the discussion of the Fig. 5/(Fig. 7 in new version) is now presented in the text (P. 11, L. 36–38, P. 12, L. 1–5).

7. Widening of the Discussion is necessary.
Our response: We have expanded the Discussions/Conclusions sections as follows: (i) the other data sets, which could be used in evaluating of the regional scale hydrological response to the expected climate changes are presented, and the corresponding references are provided (P. 13, L. 3–8), (ii) the steps of the model application for the other regions/data sets are described (P. 13, L. 9–20).

8. An improvement of the reference list is insufficient. The current foreign publications on statistics methods are suggested to be added. The reference on the archive of Bryazgin (2008) – "…personal communication…" (Page 7, Line 14) is impossible. Following further Arctic and Antarctic Research Institute is an owner of data. Check and correct this reference, please. For the Page 6, Line 9-10 is the same. The old references for the model verification \ validation (Figure 6, for example, - Ivanov and Yankina (1993)) have to be improved and current publications should be added.
Our response: The reference list was substantially improved in the revised version of the manuscript.
(i) In performing the historical context of the study, the following references were added:
1. Kritsky, S.N. and Menkel, M.F.: 1946. On the methods of studying the random variations of river flow, Gidrometeoizdat, Leningrad.
Kite, G.W. 1977: Frequency and risk analysis in hydrology. Water Resour. Publications. Colorado: Fort Collins, 224 pp.

3. Benson, M.A. 1968: Uniform flood frequency estimating methods for federal agencies. Water Resour. Res. 4, 891–908.

4. Elderton, Sir W.P, Johnson, N.L. 1969: Systems of Frequency Curves. Cambridge University Press, London, 224 pp.

(ii) In providing the reference to the source of the meteorological data used in the study:

1. Meteorological Data from the Russian Arctic 1961-2000, Version 1. 2003. Edited by V. F. Radionov and F. Fetterer. National Snow and Ice Data Center. Boulder, Colorado USA. NSIDC: National Snow and Ice Data Center. http://dx.doi.org/10.7265/N56H4FB3).

This data set was originally created basing on the data set of the Arctic and Antarctic Research Institute (St. Petersburg). The personal communication with N. Bryazgin (2008) expands the collection of the data in space and time.

(iii) In adding the recent publication connected with delineating the hydrological boundary of the Russian Arctic, the following reference was included into the list:

Nikanorov, A.M., Ivanov, V.V., and Bryzgalo V.A.: The rivers of the Russian Arctic, the current conditions under the human impact, NOC, Rostov-on-Don, 2007. (In Russian).

(iv) In performing of the recent studies, which are dedicated to the hydrological statistical and stochastic modelling issues, the following references were included:

1. Kuchment, L.S. and Gelfan, A.N.: Assessment of extreme flood characteristics based on a dynamic-stochastic model of runoff generation and the probable maximum discharge. Journal of Flood Risk Management, 4, 115–127, 2011.

2. Montanari, A. and Koutsoyiannis, D.: Modeling and mitigating natural hazards: Stationarity is immortal! Water Resour Res, 50 (12), 9748–9756, 2014.

3. Serinaldi, F. and Kilsby, C. G.: Stationarity is undead: Uncertainty dominates the distribution of extremes, Adv. Water Res., 77, 17, 2015.

9. There are a lot of abbreviations in the text without decoding or explanation. So, all names of models have to be named or the list of the used models with references can be added.

Our response: the models' abbreviations decoding and references were added to the text of the revised manuscript:

1. Roeckner, E., Bäuml, G., Bonaventura, L., Brokopf, R., Esch, M., Giorgetta M., Hagemann, S.,Kirchner, I., Kornblueh, L., Manzini, E., Rhodin, A., Schlese U., Schulzweida, U., and Tompkins, A.: The atmospheric general circulation model ECHAM5. Part I: Model description. Max Planck Institute for Meteorology Rep. 349, 2003.

2. Giorgetta, M., Jungclaus, J., Reick, C., Legutke, S., Bader, J., Böttinger, M., Brovkin, V., Crueger, T., Esch, M., Fieg, K., Glushak, K., Gayler, V., Haak, H., Hollweg, H.-D., Ilyina, T., Kinne, S., Kornblueh, L., Matei, D., Mauritsen, T., Mikolajewicz, U., Mueller, W., Notz, D., Pithan, F., Raddatz, T., Rast, S., Redler, R., Roeckner, E., Schmidt, H., Schnur, R., Segschneider, J., Six, K., Stockhause, M., Timmreck, C., Wegner, J., Widmann, H., Wieners, K.-H., Claussen, M., Marotzke, J. and Stevens, B.: Climate and carbon cycle changes from 1850 to 2100 in MPI-ESM simulations for the coupled model intercomparison project phase 5. J Adv Model Earth Sy, 5, 572-597, doi:10.1002/jame.20038, 2013.

3. Delworth, T. L., Broccoli, A. J., Rosati, A., Stouffer, R. J., Balaji, V., Beesley, J.

A., Cooke, W. F., Dixon, K. W., Dunne, J., Dunne, K. A., Durachta J. W., Findell K. L., Ginoux P., Gnanadesikan, A., Gordon, C. T., Griffies S. M., Gudgel R., Harrison M. J., Held I. M., Hemler R. S., Horowitz L. W., Klein S. A., Knutson T. R., Kushner P. J., Langenhorst A. R., Lee, H.-C., Lin S.-J., Lu J., Malyshev, S. L., Milly, P. C. D., Ramaswamy, V., Russell J., M. Schwarzkopf D., Shevliakova, E., Sirutis, J. J., Spelman, M. J., Stern W. F., Winton M., Wittenberg A. T., Wyman B., Zeng F., and Zhang R. GFDL's CM2 global coupled climate models. Part 1: Formulation and simulation characteristics, J Clim , 19 (5), 643–674, 2006.

4. Chylek, P., Li, J., Dubey, M. K., Wang, M. and Lesins, G.: Observed and model simulated 20th century Arctic temperature variability: Canadian Earth System Model CanESM2. Atmos. Chem. Phys. Discuss., 11, 22 893–22 907. 2011

5. Johns T.C., J. M. Gregory, W. J. Ingram, C. E. Johnson, A. Jones, J. A. Lowe, J. F. B. Mitchell, D. L. Roberts, B. M. H. Sexton, D. S. Stevenson, S. F. B. Tett and Woodage, M. J.: Anthropogenic climate change for 1860 to 2100 simulated with the HadCM3 model under updated emissions scenarios, Clim. Dyn 20: 583-612, 2003.

6. Collins, W.J., Bellouin N., Doutriaux-Boucher M., Gedney N., Hinton, T., Jones, C. D., Liddicoat, S., Martin G., O'Connor, F., Rae, J., Senior, C., Totterdell, I., Woodward, S., Reichler, T. and Kim J.: Evaluation of the HadGEM2 model. Met Office Hadley Centre Technical Note no. HCTN 74, 2008.

**Comments to the Abstracts**

Page 1, Line 10. "…major challenges for adaptation…". Adaptation for what? Is it regeneration or adaptation?

Our response: The challenges connected with the economic activity (the long-term development of the infrastructure) in the region are mentioned. The text was accordingly corrected (P. 1, L. 9–11).

Page 1, Line 18. Extreme flood events in the Russian Arctic are connected only with spring snow melting seldom. The most hazard events are occurred during the multiplying of a river flood, tides and surges on the Arctic coast (marsh areas). It can effect on a river discharge more than 50-70 km upstream of rivers. There is no explanation of such event in the article as well as in the results of modeling.

Our response: In this study, the regional scale assessment of the extreme flood events was performed based on the observations for the catchments of medium size (from 1,000 to 50,000 km$^2$), which are located in single climate zone. We do not consider the features of the runoff processes in the local scale (appeared on the small watersheds) and in the global scale (revealed on the huge watersheds located within several climate zones). The flooding due to ice jams and tides/surges were not elaborated. This explanation now is added to the revised text (P. 9, L. 29–34).

Page 1, Line 20. Abbreviations in abstract should be interpreted before, not in the text.

Our response: The text of abstract was corrected and the abbreviations were excluded.

**Comments to the Introduction.**

Page 2, Line 13-14. The models can be global or regional but stochastic of

physically-based etc. What exactly models had been used in the paper?

Our response: The "in-house" developed probabilistic model described in Kovalenko (1993, 2014) were used in performing the regional scale assessment of the parameters of PDFs of the spring flood flow depth of runoff. The basic principles and hypotheses behind this modelling approach are shortly presented in the Annex of the revised manuscript.

Why the Markov randomisation had been used if the authors have a huge amount of observed data?

Our response: The simple Markov chain is the basic paradigm behind the traditional flood frequency analysis, which is used in the engineering hydrological applications (see e.g. Bulletin 17-B or Rogdestvenskiy, 1988). This model was proved by the statistical analysis of the autocorrelation functions, which were obtained based on the numerous time series of annual and spring flood runoff (Rogdestvenskiy, 1988). We add this comment to the revised text (P. 3, L. 22–25).

Authors said about the cheapest (Page 3, Line 2) stochastic approaches in the comparison of physically-based. But using only 3 parameters of PDF (Page 3, Line 3-4) for meteorological variability is also insufficient for the whole Arctic climate prediction.

Our response: The benefit of the method used this study is in skipping the simulation of the future hydrological time series (Fig. 1 in the revised manuscript). Within this study we do not predict the meteorological variability and climate for the whole Arctic.

Page 3, Line 5. Kovalenko is not the first scientist who suggests the stochastic approach for hydrological engineering.

Our response: The historical aspect of the method is not presented in the text with corresponding references (P. 3, L. 20–23).

Page 3. There is no explanation of stationary and quasi-stationary regimes.

Our response: The stationary regime means that the statistical parameters of runoff PDF are not changed for past and future time periods (as considered in classical engineering application). The quasi-stationary regime means that these parameters of PDF are differ for past and future time periods. The explanation is now given in the text (P. 3, L. 31–36).

Page 3, Line 14. I am interested how the authors explain of using the same approach for drought extremes in European part of the Arctic as well as for extreme flood events in the West and East Siberia, - such climatically and physic-geographical different regions.

Our response: The approach proposed based on the theory of Markov processes, which can be apply to evaluate the extremes from the PDF. In engineering hydrology the extremes (floods and droughts) are defined as the PDF tailed values, which conform to given probability of exceedance (0.1, 1, 5, 10 % for floods and 90, 95, 99, 99.9 % for the droughts). Then, the issue of the floods and droughts prediction is only the estimation of the PDF parameters, which are expected under a new climate. The geographical peculiarities of the regions are accounted by the parameters of the model (2), thus the regional-oriented parameterization scheme is usually required. We add this explanation into the text of revised manuscript.

Page 3, Line 27. What "domains" does the authors mean? They had not been explained before.

Our response: The "domain" means geographical region, the territories with specific climate conditions, land cover and runoff regime. We replace this word with the word "territories" in the revised manuscript.

**Comments to the Methods and data**

The chapter should be restricted totally. The methods are not clearly outlined. Some comments should be added to the chapter.

Our response: This chapter was improved in the revised version, in particular, we (i) discussed the opposing views on the significance of affecting the climate change to the hydrological regime (P. 2, L. 3–13); (ii) provide the background, why there needed to be a statistically significant shift in the observed time series to perform the model cross-validation (P. 6, L. 22–29); (iii) discussed the dataset used in the study and references connected with data (P. 8, L. 10–13, P. 13, L. 3–8); (iv) explained the case of "no model" represents in the cross-validation section (P. 8 L. 28–31); (v) added description of the data sets used in the study (Fig. 4, P. 9, L. 5–13); (vii) presented the Fig. 4, which shows the data sets used; (viii) improve the list of references.

There are results (for example, for the Nadym River) that could be removed to the result and discussion chapter.

Our response: There are no results for the Nadym River, which are shown in the method and data section. The result of the cross-validation for the Yana River shows the successful / non-successful the cases of the "nominal" prediction of the PDFs. It is necessary in this section from our point of view.

The list of equations and their conventional signs is recommended to be done. Some equations are not used in following text but another are written twice (Cv, for example), the equation GN (Page 4, Line 30) does not have a number, etc.

Our response: Now the duplicated equation was removed, and the Annex with basis of the approach was add to the revised manuscript. Only the equations, which have references in the text were numbered.

Are there differences between Cv and Cvf, Cs and Csf equations?

Our response: The index "f" (Cvf) indicates that the coefficient of variation is calculated for the future time period based on the modeled two statistical moments. The equation used to calculate the CVf is similar as for Cv and it is provided now in the annex of the manuscript. Thus we exclude this equation from this section and include it to the Annex.

Are the authors sure that Cv\Cs will be constant (Page 5, Line 8)?

Our response: In present study we assume, that the ration of Cv\Cs is constant for the past and future. However, the assumption of the constant ratio of Cv\Cs for the past and future time periods can be refused in the future study. Then, the system of the equations for three statistical moments (A.5) should be used (see Annex Eq. A.5) and Cs can be evaluated from three statistical moments.

The list of all parameters from the calculated according to SP equation (Page

3, Line 45) for the estimating rivers is recommended to be added to the text.

Our response: In performing of cross-validation of the model (2) we did not used the extremal discharges with low probability of exceedance and the nominally predicted and empirical PDF are compared integrally by the goodness-of-fit statistical tests. Thus, there is no needs to calculate the maximal discharges using Eq. (1), and they are not shown in the text. However, the values of the parameters in Eq. (1) for the Nadym River is now presented in the text (P. 12, L. 23–25).

The reason of a runoff reduction (Page 4, Line 5) and using b and n factors and degree have to be also explained.

Our response: Eq. (1) includes parameters $b$, which is the additional area which adjusts the reduction of the runoff (km$^2$) and $n$, which is degree of a runoff reduction. The numerical values of these parameters are presented in the look-up tables e.g. Guidelines SP33-101-33 (2004) or (1984).

The "...flood flow depth of runoff..." (Page 4, Line 8) and "...the spring flood depth..." (Page 4, Line 23 and following the text) is misunderstanding. Are these same things or differences? If it is the same it can be unified in the text.

Our response: In this study, these two terms mean the same, and we use "spring flood flow depth of runoff" in the text of revised manuscript.

The authors supposed "...the future time period 2010-2039..." (Page 4, Line 12) in spite of that a current time is 2016. Have the authors done the verification of their model for 2010-2015? What the result have they received? May be it can help in understanding of the model availability for a runoff prediction in the Arctic.

Our response: The model (2) allows estimating the multi-year mean value and coefficient of variation of the spring flood flow depth of runoff. In estimation of these values, the 5-year period is not enough. This is the reason why the model (2) can not be verified using the observations for 2010-2015.

How the authors estimate the reference periods (Page 4, Line 38)? Why it is necessary for the methods and modeling? It is unclear in the chapter.

Our response: The reference period is the time slice with (i) the observed data available and (ii) steady climate and runoff regime. The "steady" means that there are no statistically significant trends and changes in the mean values of meteorological and hydrological characteristics. The reference period is necessary for the modeling since (i) the parameters of the model (2) are evaluated basing on the climatology and runoff statistics of this period, and (2) the warning regions are delineate basing on the differences in the mean values and coefficients of variation for the reference and projected periods. We add the explanations to the text of revised manuscript (P. 5, L. 23–30).

Is the sub-periods in the table 1 and 2 (and Page 6, Line 7 and Line 39, for example) are the same as "reference periods" or not?

Our response: In the cross-validation section (and Tables 1 and 2) we used terms "training" and "control" periods, the text of the revised manuscript was corrected.

What the differences \ similarities between "reference periods" and "training" and "control" (Page 6, Line 28, 30, 38) or "...reference and future ..." (Page 5,

Line 8) periods.

Our response: The model (2) operates within two time periods with steady climate and runoff regime (idea of quasi-stationarity). One time period is used to evaluate the model parameters, and it is noticed as "training" period in the verification section and "reference" in the section of "data and method". Other time period is the period of the prediction (or the nominal prediction), and it is noticed as "control" period in the verification section and "projected/future" in the section of "data and method".

Later, in the table 6, the "Historical period" is. There are misunderstanding of periods definitions in the text.

Our response: The "Historical period" was replaced by the "Period 1950–1980" in the text of the revised manuscript.

The sub-periods were selected according to the statistically significant differences in the first statistical moments (Page 5, Line8-9) but authors have not explained what does "…subsampled mean values…" \ "..subsample equals…" means.

Our response: The "subsample" is the observed time series within the selected sub-period, it is used to evaluate the mean values and coefficient of variation (or the first and the second statistical moments).

Dimensions of the first ($m_1$ (mm)) and the second ($m_2$ (mm$^2$)) statistical moments of the spring depth of runoff (Page 4, Line 25-26) could be explained too as well as the parameter GN (mm$^2$). How these two statistical moments have been received (see the table 1 and table 2)?

Our response: The the statistical moments of the spring flood flow depths of runoff are estimated from the observed time series using the method of moments (Bowman and Shenton, 1998). The dimension of the first statistical moment ($m_1$ or the mean values) is equal to the dimension of the value (for the spring flood flow depth of runoff the dimension is mm, since this value is calculated as the volume of spring flood flow (m$^3$) from the drainage basin divided by its area (m$^2$)). The dimension of the second statistical moment ($m_2$ or dispersion) is equal to dimension of random variable square. The parameter GN reflects the dispersion of the precipitation, and the dimension of this parameter is mm$^2$.

The reference for the "…Pearson chi-squared and Kolmogorov-Smirnov one-sample tests…" is necessary as well as the explanation for the used methods.

Our response: The following reference is add to the list: Hollander, M., Wolfe, D.A. and Chicken, E.: Nonparametric statistical methods, 3d edition, Wiley, 848 p., 2014.

What does "…cross-validation…" mean (Page 6, Line 16)?

Our response: The cross-validation is a model evaluation method, which allows performing the model ability to reproduce the measurements. In simplest case, the dataset of the measurements (observations) is separated into two sub-sets, called the training set and the testing/control set. Then, the training set is used to evaluate the model parameters, which are further used to calculate the modelling (or nominally predicted) dataset to compare with the testing/control set using chosen measure (the statistical goodness-of-fit tests in our case). The procedure of defining the training set and testing/control set

are described in the section 2.2 together with the results of the ability of the model suggested to represent the empirical PDFs (P. 6, L. 22–27).

There are no data in the chapter 2.3. Rename it or add explanation which exactly data had been used. Is it measured meteodata or received data from the climate models? One part of the paper is about analyses of measured multi-year data from Russian meteostations (Water Cadastr (Page 6, Line 41-44)), the second part is about modeled climatic data. The connection between these two parts is incoherent shown in the methods chapter.
Our response: The explanation about the data sets used in the study was added to the revised text (Fig. 4 , P. 9, L. 2–7).

According to the reference on Page 6, Line 5-6 the "…node the mean values and the coefficients of variation of the spring flood depth of runoff were extracted from the maps…" that had consequently been built before 1986. The 80th of last century is a time of the beginning of a huge climatic change in the Arctic. Do the authors suppose that all coefficients are the same and can be used for modeling and prediction?
Our response: The changes in the observed meteorological and runoff characteristics after 1980s are important in choosing of the reference period. In this study, the assumption of the quasi-stationarity of the changes in the climate and runoff regime was used. It means, that there are two periods with steady climate (defined by mean values of meteorological characteristics) and runoff regime (defined by mean values, coefficient of variation and coefficient of skewness for runoff characteristics). But, the statistical values are different for these two periods. Then, the reference period have to be defined as time slice without any statistically significant trends in climatology and runoff characteristics. This is the reason why we do not consider the runoff data on the prediction stage for the regional scale hydrological projections.
However, on the model verification stage (cross-validation) we used the observations until 2006 for the catchments, where two quasi-stationary periods were found.

On the base of a climate change the Cv\Cs ratio has to be also changed (Page 8, Line 16) and cannot be used for modeling as a fixed coefficient. If the Cv\Cs ratio is equal it means that Cv or Cs coefficients have simultaneous trends that need to be explained additionally.
Our response: In this study we used the assumption of constant ratio of Cs/Cv, however, this assumption can be avoided in future studies. The system of equations for three statistical moments (A5) have to be used in this case.

**Comments to the Result and discussion**
There are some examples from the global models calculations and predictions in the chapters. Authors' received results presented in the tables 1-6 are without properly clarification and interpretation in the Result and Discussion chapter.
Our response: In revised manuscript the discussions were expanded for the results, which are presented in the tables and figures. Also, the historical context and special questions connected with the method used are clarified.

How do the authors understand "…alarm regions…" (Page 11, Line 31-32)? Discussion on the Arctic alarm-regions analyze could fulfill the text and

conclusion chapters.

Our response: The term of "alarm" was replaced by "warning" regions. The regions, where the PDF tailed values with low probability of exceedance are going to change substantially are defined as "warning regions".

Unfortunately, authors did not compare their results with other publication sufficiently.

Our response: It is very difficult to compare our results with other studies, since the different hydrological characteristics are projected. In revised text we expand the discussion of this issue.

**Small marks to the text**

Page 3, Line 45 and Page 4, Line 6. The abbreviation should be clarified.

Our response: The "SP" is not abbreviation, this is the index of the state guidelines.

Page 4, Line 1. How the authors "…a flood coincidence factor…" had determined?

Our response: The a flood coincidence factor reflects the water income to the catchment (due to melting), which affect to the shape of hydrograph. It is usually depend on the geographical regions and obtained from the look-up tables (P. 4 L. 36–37).

Page 4, Line 2. The dimension of "probability" have to be marked and type of "… an exceedance probability curve…" should be also noticed.

Our response: It was corrected.

Page 2, Line 34. The name of "…Lehner…" – check the format, please.

Our response: It was corrected.

Page 2, Line 36. GCMs – decode the abbreviation in advance, please.

Our response: It was corrected.

Page7, Line 1. Not "…Arctic…" but "the Arctic".

Our response: It was corrected.

**Comments to the Supplementary materials**

For all tables: in the title of a table all denotations have to be explained.

Our response: The notations are now presented in the revised manuscript.

Table 1. Rives can be divided to a large \ middle \ small size of their catchments.

Our response: All rivers in Table 1 have the catchments of middle size (according to definition given in the Guideline: Hydrology of land,Terms and definitions, Moscow, 1988).

Table 2. Why are there two periods for each river? Could be periods marked as "training" and "control"? The differences of the period lengths for each river are not clear explained in the text.

Our response: Both periods were used in cross-validation, then they are both "training" and "control" (since the validation was done forward and backward). The periods lengths depend on the year having the value of $t$-test exceeding the critical value 0.05 level of statistical significance.

Table 4. "…Reference climatology…" is misunderstanding.

Our response: The climatology for the period since early 1930s till 1980 which was considered as a reference in our study (P.   L.   ).

Table 6. What does the "…Historical period…" mean?

Our response: The period was specified in the revised text.

Figures 3. Are the observed data in the figures for all period of measurements? Did the mean value mark on a screen?

Our response: The mean values/coefficients of variation are given for the reference period (1930–1980). The figure was corrected.

Figure 5. The resolution of the pictures are not enough for good understanding and comparison.

Our response: The figure was corrected.

Figure 6. The data of discharge for the presented models are very variable. It is recommended to be discussed in the text.

Our response: The discussion about results was extended.

We thank the Reviewer for the questions, useful comments and suggestions, which have allowed to clarify and improve the text of the manuscript.

---

## Author Response (AR1)

Response to the Handling Editor

We thank the Editor and the for the comments and suggestions. They were very useful in further improving of the manuscript. The Editor's comments are copied below, and our responses are written after each comment.

Concerning the manuscript hess-2015-504 titled "Assessment of extreme flood events in changing climate for a long-term planning of socio-economic infrastructure in the Russian Arctic". We have provided two review reports about this manuscript. Both the two reviews highlight the need of a revision of the manuscript, and I agree with them. Thus I would like to give you the opportunity to submit a revised version of the manuscript, which must clarify the issues raised by the Reviewers.

In the specific, I would like to ask you to
- clarify the methodology proposed. For example, something is wrong in Eq.(1);
The explanation of the notations used in the Eq. 1 is extended to fix the mismatches in the units (P. 5 L. 3), which was appeared in the text of the discussion manuscript.
- clarify the advances respect the existing methodologies;
To clarify the peculiarity and advances of the suggested method comparing with the existing methodologies, the explanation were extended in the text (P. 2, L. 28–38, P. 3, L. 1–19 and P. 4, L. 1–6) and the Figure was added (Fig. 1). Also the historical context of the study was also extended (P. 3 L. 20–31), and the relevant references are provided.
- explain/give details about the possibility to apply the methodology to other data sources;
The applied method does not restricted by the specific province and dataset, and it also can be used to perform the assessment of the parameters of the PDF for other regions. Among others, the following datasets can be used (i) the Global Runoff Data Centre, Germany; (ii) the Environmental Information System (HERTTA), Finnish Environment Institute; Vatten Webb, Swedish Meteorological and Hydrological Institute. The details about the possibility to apply the methodology to other data sources now are given in the conclusions together with the description of the steps how these data can be utilized to perform the modelling (P. 13, L. 3–28).
- include discussion about the uncertainty;
The discussion about the uncertainty of future parameters of PDF and tailed values is now included (P. 10, L. 11–16).
- fix\discuss the introduction according to the suggestions given by Dr. Serinaldi.
We have discussed the opposing views on the significance of affecting the climate change to the hydrological regime (P. 2, L. 3–13).
The corrections were also added to the text following to the comments of the reviewers:
1. the background, why there needed to be a statistically significant shift in the historical time series to perform the model cross-validation (P. 6, L. 22–29).
2. the discussions about dataset used in the study and references connected with data (P. 8, L. 10–13, P. 13, L. 4–9).
3. the explanation what the case of "no model" represents in the cross-validation section (P. 8 L. 28–31).
4. the data sets used in the study (Fig. 4, P. 9, L. 5–13).
5 the reasons for selecting particular climate models (P. 9, L. 20–23).
6. the degree in which the results of the study were already published (P. 10, L. 24–26).

Additionally, the tables and figures were improved, and names of the geographic domains were corrected. The language issues were also checked. We also provided the detailed responses to the comments of the reviewers as the separate papers.

[revised manuscript text omitted]

---

## Referee Report (RR1)

**Assessment of extreme flood events in changing climate for a long-term planning of socio-economic infrastructure in the Russian Arctic**

**By E. Shevnina, E. Kurzeneva, V. Kovalenko, and T. Vihma**

**Submitted to Hydrol. Earth Syst. Sci. Discuss.**
*MS-NR: hess-2015-504_R1*
* * *
**REFEREE REPORT**

**General comments**

First of all I would like to reply to Authors' responses in order to better clarify some technical points and my point of view.

Uncertainty: confidence intervals for return periods I mentioned in my report are based on basic results for order statistics, which approximate confidence interval for quantiles. These issues are well-known and already discussed many years ago by Vit Klemes, among others. They can be found in his papers as well as books on applied statistics.

Saying that confidence intervals cannot be build (while credible intervals can) makes little sense. First of all, in my report I specified "confidence intervals or something similar"; so, if you prefer (Bayesian) credible intervals, or Dempster's imprecise probability or whatever else, you are free to implement it; in any case, given the limited information (data) used, uncertainty should be shown in some way because the comparison of point estimates is not enough and gives a false sense of accuracy. Second, I gave a "sharp" description of the method because the rationale is the same as whatever approach introducing dynamically-varying distributions where the parameters change according to covariates: you derived the relationships between runoff PDF parameters and rainfall moments via stochastic differential equations, while they are derived more commonly using pure empirical/statistical approaches. However, in both cases, uncertainty and CIs can be quantified irrespective of the availability of future data (which are not available by definition). In this respect, I suggested simple bootstrap techniques: leaving aside the uncertainty of future covariates (rainfall and temperature), the sampling uncertainty can easily be quantified by bootstrapping B times the observed records and re-estimating the model for the period of records, and then projecting this bundle of distributions by applying the proposed method to each of them. This way, we obtain a set of future distributions (driven by the same meteorological forcings, which are assumed to

be uncertainty-free) describing the propagation of the sampling uncertainty (which affects the parent distribution estimated in the period of records). Of course this is only one source of uncertainty but often is the most relevant. This technique, as well as more refined ones, is fully general and can be applied for whatever model; thus sentences such as "In this case, the classical estimates of the uncertainties on the PDF tailed values (i.e the confidence intervals) can not be applied since the future runoff time series do no exist." makes little sense. As mentioned above, the only difference between this paper and others dealing with nonstationary distributions driven by covariates is the derivation of the relationship between PDF parameters and covariates (by (simplified) physical arguments rather than empirical relationships); however, in both cases the same inferential results (such as uncertainty assessment) can be applied. Unfortunately, there is a general tendency to falling in love with a particular method missing its analogies and relationships with other techniques missing the more general picture. This prevents to recognize that a method such as that proposed in this study is only a particular case of techniques already used and equipped with a set of tool which can be applied also in this case. So, please, perform some simple bootstrap exercise and show at least sampling uncertainty effects.

Bullettin 17b: The Authors's answer does not match my comments; there should be a misunderstanding. Actually, I did not say that regionalization is anachronistic, but that the at-site analyses based on short time series are unreliable and somewhat anachronistic; the point is that when I talk about regionalization I mean techniques such as index-flow method where data from multiple sites are merged to form a unique longer sample under the hypothesis that spatial information can replace temporal information. "Regionalization" is different from "regional analysis" (i.e. visualization of at-site/local variability across an area), which is actually what is done in this study. In this respect, the study is somewhat anachronistic because overlooks the widely recognized unreliability of at-site estimates and omits a fair communication of the uncertainty (I mean the propagation of the sampling uncertainty affecting the PDF of the period of records into the future projections).

Nonstationarity and uncertainty: Again, I understand the Authors' reply (which is quite common) but the point is different: using nonstationary models implies the identification of a deterministic trend (predictable with negligible uncertainty over a time window of interest), which in turn requires a deterministic attribution, and this is not the case in complex systems such as global climate dynamics. So, in this respect, dynamically-varying distributions provide pictures of "what if" scenarios under some given conditions (e.g. emissions) which are deterministic because they are imposed in the climate models simulations. So, my comments do not refer to nonstationary methods by themselves, but on their general use taking for granted the presence of "deterministic trends" that are often only stochastic and related to long range fluctuations of stationary processes.

Skepticism: it is related to the arguments above. Skepticism comes before hypotheses and axioms, as it has an epistemological role, meaning that it provides a rule to assess the suitability of hypotheses and axioms in order to retain sound hypotheses and discard

nonsense/wrong assumptions. Skepticism is exactly the rule that helps quicker advances avoiding nonsense theories and starting points leading forward but in the wrong direction.

Concerning my last remark in the previous report, the answer is vague and no very convincing. Every method can provide a forecast: stationary methods provide stationary forecasts, nonstationary methods nonstationary forecasts. For design purposes, the point is which one (if any) is the most credible and reliable based on the available information, in order to make a decision. Every method is useful in principle for practical applications, but it depends on whether it fits the problem at hand or it is used under wrong hypotheses. Again, in the present context, for practical applications, uncertainty and reliability should be assessed.

**Specific comments**

Although the Authors state that "the language issues were also checked", actually it seems that they did not, as the revised text shows several syntax/grammar errors as well as the use of inappropriate terminology, even in the abstract (e.g. "probability dencity functions", "correction shold be applied"). In general, almost all the paragraphs introduced in the new version show some error. In the following, I report only some examples as the manuscript requires a professional proofreading.

L3-13: This new paragraph does not describe correctly the actual situation. The point is not if changes in climate drive changes in runoff, as this is obvious. The point is that nonstationary models require that we are sure about the future changes or, in other words, that the changes are deterministic (predictable). If the deterministic evolution (nonstationarity) of the process is only hypothetical, it is evident that we have no idea of how it will evolve, and supposed trends or regime shifts can simply be local fluctuations of perfectly stationary processes. In other words, we have hypothetical projections, but we have no idea of the actual evolution of climate and so runoff. Therefore, tools for nonstationarity are surely interesting, but their suitability in this context is highly questionable because they involve an additional source of uncertainty related to the unknown evolution pattern. This stresses once again the importance of assessing the uncertainty and reliability of design values.
"We consider it improbable that changes"… do you mean something like "We consider that it is improbable that changes"?? Please reword.

P3L4: please consider "physically-based model described by dynamic equations"

P3L4: please consider "the meteorological signal could be simulated by random generators." Do you mean "weather generators"? By the way, as I can see "a-priory" several time throughout the text, it is worth recalling that it should be "a priori": it is Latin and means "in (a) advance (priori)". If you are not familiar with such a kind of expressions please avoid them and use plain English.
P3L14-15: "the parameters of PDF are directly *simulated* from the meteorological mean values" please use correct terms: simulated means generate by e.g. a MC procedure. In

this case, parameters are estimated as functions of covariates (meteorological mean values). Estimation and simulation have a different meaning and should be used in the correct context. "These parameters are further used to model PDF with theoretical distribution"??

P3L17: please replace "detrimental" with "extreme"

P3L34: "classical assumption used *behing* the *engeenering* applications" please consider "classical assumption used in engineering applications"

P4L34: "is calculated according method from"… do you mean "is calculated according to the method proposed (discussed) by (in)"

P4L36: Please consider "which reflects the water income to the catchment (due to melting) that affects the shape of hydrograph"

P5L23: "based on"

P5L28-30: what do you mean? Please reword in a more readable way

P6L21-22: "different statistically significant PDF parameter values" do you mean that the difference between the parameter values in the two periods is statistically significant?

P6L25: "which allows performing the model ability to reproduce the measurements. In simplest case…" -> "which allows one to assess the model ability to reproduce the measurements. In the simplest case…"

P6L27-29: "Then, the training set is used to evaluate the model parameters, which are further used to calculate the modelling (or nominally predicted) dataset to compare with the testing/control set using chosen measure (the statistical goodness-of-fit tests in our case)." What does it mean "to calculate the modelling dataset"? Please reword using appropriate terms to describe corresponding concepts.

P9L20: Chylek et al. (2011) was not accepted for final publication. Please provide a published reference.

P9L22: "the the"

P10L13: "do no"

P10L26: do you mean "especially"? Please reword the sentence.

P10L26: "to calculate the maximal runoff from tailed values with required probability of exceedance" -> "to calculate the yearly maximum runoff with required probability of exceedance". Please use homogenous terms to describe the same object, avoiding e.g. "maximal runoff" for what is previously defined as "yearly maximum runoff" and brand

new terms like "tailed values" or "maximal extremes" to denote "extreme quantiles" or simply "extreme values".

P13L18: "toevaluate"

Sincerely,

Francesco Serinaldi

---

## Referee Report (RR3)

**Final review on "Assessment of extreme flood events in changing climate for a long-term planning of socio-economic infrastructure in the Russian Arctic" by E. Shevnina et al.**

The final version of the paper "Assessment of extreme flood events in changing climate for a long-term planning of socio-economic infrastructure in the Russian Arctic" by E. Shevnina et al. can be fully accepted to the HESS. During several iterations of the paper improvement the methods and results interpretation have been considerably enhanced. Current explanations of using and received data are clarified, figures are demonstrative and text is readable. Authors, according to the climate scenarios, show a new methodic for a social-economical prediction quite well. Different variants of environment condition change and the Arctic river runoff affect are explained detail with applying purpose.

However, authorial methods can be further discussed but is useful anyway. Lack of the East Siberia runoff and meteo data are regrettable, it does not give a possibility to analyze current and to predict future hydrological conditions in this huge region but has the place to be. Most part of references for methods explanation are only in Russian. Hopefully, the reviewer is Russian too and can read it easy but it is not suitable for other foreign readers. Preparing of a target methodical paper(s) with an explanation and a clarification of several all-Russian common methods are recommended for an intended auditorium.

This paper will be fruitful for hydrologist by the presented data, the method, and results mapping and predictions.

Dr. Irina Fedorova

---

## Author Response (AR2)

Assessment of extreme flood events in changing climate for a long-term planning of socio-economic infrastructure in the Russian Arctic
By E. Shevnina, E. Kurzeneva, V. Kovalenko, and T. Vihma

Submitted to Hydrol. Earth Syst. Sci. Discuss.
MS-NR: hess-2015-504_R1
* * *
RESPONSE TO THE REFEREE REPORT by Francesco Serinaldi
General comments
We thank Francesco Serinaldi for his useful comments. Below we provide our responses (blue text) to his comments and questions (black text).

First of all I would like to reply to Authors' responses in order to better clarify some technical points and my point of view.
Uncertainty: confidence intervals for return periods I mentioned in my report are based on basic results for order statistics, which approximate confidence interval for quantiles. These issues are well-known and already discussed many years ago by Vit Klemes, among others. They can be found in his papers as well as books on applied statistics. Saying that confidence intervals cannot be build (while credible intervals can) makes little sense. First of all, in my report I specified "confidence intervals or something similar"; so, if you prefer (Bayesian) credible intervals, or Dempster's imprecise probability or whatever else, you are free to implement it; in any case, given the limited information (data) used, uncertainty should be shown in some way because the comparison of point estimates is not enough and gives a false sense of accuracy.

Our response: There are several sources of the uncertainties in the method described in our manuscript:
1. from the assumed (given a priori) type of distribution (Pearson type III);
2. from the limited length of hydrological time series, which were used to evaluate the parameters of distribution for the reference period.
3. from the limited length of meteorological time series to evaluate the climatology for the model's parameterization.
4. from the uncertainties in future climate, as projected applying climate models (forcing), which are imperfect;
5. from the mapping errors due to the interpolation technique;
6. from the errors due to the calculation of maximal discharges from spring flood depth of runoff (Eq. 1).
In our study, the problem of the uncertainties should be considered for two cases: (i) the point estimates at a particular site (the Nadym River at Nadym City, Fig. 8) and (ii) the maps of the regions with substantial changes (Fig. 7). In both cases the uncertainties come from all items of the list above. For the estimates at a site with observations, the uncertainties from item 2 were evaluated and presented further as our response to the second general comment of the Reviewer. However, we cannot apply the same method to the maps since it needs additional assumptions and requires further studies. We discuss the uncertainties of the maps on page 10, lines 20–26, and the method used to outline the regions on Fig. 7. More accurate estimation of the uncertainties is a topic of further studies.
In the revised version of the manuscript we write as follows:
*p. 11, l. 27-38*: There are several sources of uncertainties in the method described above: (1) from the

assumed (given a priori) type of distribution (Pearson type III); (2) from the limited length of hydrological timeseries, which were used to evaluate the parameters of the distribution for the reference period; (3) from the limited length of meteorological timeseries to evaluate the climatology for the model's parameterization; (4) from the uncertainties in future climatology provided by climate models (forcing); (5) from the mapping errors due to interpolation techniques; (6) from the errors due to the calculation of the maximal discharges from the spring flood depth of runoff (Eq. 1). The uncertainties inherent to the simulated PDFs' parameters include items 1–5 above. These uncertainties are evaluated by Kovalenko (1993) for the maps of means/$C_v$, provided by Pogdestvenskiy (1986) and Vodogretskiy (1986) with an assumption that the errors in the future and past climatology are the same. The average percentage errors in the projected means/$C_v$ equal to 15 % / 25 %, thus it is suggested to consider the changes in the PDFs' parameters to be substantial, if they exceed the reference values by more than these thresholds. Then the regions with substantial changes in the means and $C_v$ of the spring flood flow depth were outlined (Fig. 7).

Second, I gave a "sharp" description of the method because the rationale is the same as whatever approach introducing dynamically-varying distributions where the parameters change according to covariates: you derived the relationships between runoff PDF parameters and rainfall moments via stochastic differential equations, while they are derived more commonly using pure empirical/statistical approaches. However, in both cases, uncertainty and CIs can be quantified irrespective of the availability of future data (which are not available by definition). In this respect, I suggested simple bootstrap techniques: leaving aside the uncertainty of future covariates (rainfall and temperature), the sampling uncertainty can easily be quantified by bootstrapping B times the observed records and re-estimating the model for the period of records, and then projecting this bundle of distributions by applying the proposed method to each of them. This way, we obtain a set of future distributions (driven by the same meteorological forcings, which are assumed to be uncertainty-free) describing the propagation of the sampling uncertainty (which affects the parent distribution estimated in the period of records). Of course this is only one source of uncertainty but often is the most relevant. This technique, as well as more refined ones, is fully general and can be applied for whatever model; thus sentences such as "In this case, the classical estimates of the uncertainties on the PDF tailed values (i.e the confidence intervals) can not be applied since the future runoff time series do no exist." makes little sense. As mentioned above, the only difference between this paper and others dealing with nonstationary distributions driven by covariates is the derivation of the relationship between PDF parameters and covariates (by (simplified) physical arguments rather than empirical relationships); however, in both cases the same inferential results (such as uncertainty assessment) can be applied. Unfortunately, there is a general tendency to falling in love with a particular method missing its analogies and relationships with other techniques missing the more general picture. This prevents to recognize that a method such as that proposed in this study is only a particular case of techniques already used and equipped with a set of tool which can be applied also in this case. So, please, perform some simple bootstrap exercise and show at least sampling uncertainty effects.

Our response: for the point estimate (future maximal discharges with particular exceedance probability) the uncertainty described in the items 3-6 of the list above cannot be estimated quantitatively, at least at the moment. The uncertainties described in item 1 of the list might be evaluated with creditable intervals, but applying this method in hydrology requires an extensive specific study. We estimate the uncertainties described in item 2 of the list for the Nadym River at Nadym City.

In the revised version of the manuscript we write as follows:

*p. 12, l. 24-30*: The confidence intervals for the reference values of $h_{1\%}$ were calculated using the formulas suggested by Ashkar and Bobée (1988) in assumption that the given distribution is Pearson

type III. The 90% confidence interval for the reference $h_{1\%}$ equal to ±64.5 mm, which is about 23 % of the quantile value. The projected values of $h_{1\%}$ are within these uncertainties for all considered climate scenarios (Table 6), thus due to the short time series we can't prove that the future changes in $h_{1\%}$ are statistically significant. However, we suggest to take into account the projected climatology in calculation of hydrological risks because of practical reasons: it is better to prevent an accident rather than to deal with its consequences, which may be more expensive than the initial investment (Räisänen and Palmer, 2001).

*p. 16, l. 19-20:* Ashkar, F. and Bobée, B. , Confidence intervals for flood events under a Pearson 3 or log Pearson 3 distribution. Journal of the American Water Resources Association, 24: 639–650. doi:10.1111/j.1752-1688.1988.tb00916.x, 1988.

Bullettin 17b: The Authors's answer does not match my comments; there should be a misunderstanding. Actually, I did not say that regionalization is anachronistic, but that the at-site analyses based on short time series are unreliable and somewhat anachronistic; the point is that when I talk about regionalization I mean techniques such as index-flow method where data from multiple sites are merged to form a unique longer sample under the hypothesis that spatial information can replace temporal information. "Regionalization" is different from "regional analysis" (i.e. visualization of at-site/local variability across an area), which is actually what is done in this study. In this respect, the study is somewhat anachronistic because overlooks the widely recognized unreliability of at-site estimates and omits a fair communication of the uncertainty (I mean the propagation of the sampling uncertainty affecting the PDF of the period of records into the future projections).

Our response: the Bulletin 17b was mentioned in our study as  an example of the guideline for hydrologists, which: (i) gives the strict recommendation to apply an assumed (given a priori) type of distribution for the PDFs; (ii) provides the method to evaluate the PDF parameters from the observations; (iii) recommends to map the multi-year runoff statistics. The aim was to stress that in our study we also use the techniques of engineering hydrology. In this study we rely on the instructions from SP33-101-2003 (Russian analogue of Bulletin 17b) in applying the Pearson type III distribution to model the PDFs of the spring flood depth of runoff. This was mentioned in the previous version of the manuscript on *p. 1, l. 38, p. 2, l.1.* These PDFs are usually modelled with three parametric distributions (e.g. Pearson type III or Log Pearson type III) using mean value, the coefficient of variation and coefficient of skewness (SP33-101-2003, 2004; Bulletin 17–B, 1988).

Nonstationarity and uncertainty: Again, I understand the Authors' reply (which is quite common) but the point is different: using nonstationary models implies the identification of a deterministic trend (predictable with negligible uncertainty over a time window of interest), which in turn requires a deterministic attribution, and this is not the case in complex systems such as global climate dynamics. So, in this respect, dynamically-varying distributions provide pictures of "what if" scenarios under some given conditions (e.g. emissions) which are deterministic because they are imposed in the climate models simulations. So, my comments do not refer to nonstationary methods by themselves, but on their general use taking for granted the presence of "deterministic trends" that are often only stochastic and related to long range fluctuations of stationary processes.
Skepticism: it is related to the arguments above. Skepticism comes before hypotheses and axioms, as it has an epistemological role, meaning that it provides a rule to assess the suitability of hypotheses and axioms in order to retain sound hypotheses and discard nonsense/wrong assumptions. Skepticism is exactly the rule that helps quicker advances avoiding nonsense theories and starting points leading forward but in the wrong direction.
Concerning my last remark in the previous report, the answer is vague and no very convincing.

Every method can provide a forecast: stationary methods provide stationary forecasts, nonstationary methods nonstationary forecasts. For design purposes, the point is which one (if any) is the most credible and reliable based on the available information, in order to make a decision. Every method is useful in principle for practical applications, but it depends on whether it fits the problem at hand or it is used under wrong hypotheses. Again, in the present context, for practical applications, uncertainty and reliability should be assessed.

Our response: the uncertainties were evaluated for the point and maps in the revised manuscript. We recognized that other uncertainties cannot be evaluated by existing methods without extensive further studies. We also recognize that our estimates of future risks, although differ much from the current values, lay within the confidence intervals. However, usually any statistical results depend on the sample volume in a formal way: when we have a small sample, we might get even large changes to be insignificant. And on the contrary, when we have a very large sample, we might get the statistical significance even for the very small changes. It is our scientific intuition and practical common sense, how we treat the situation. That is why we propose to account for the future climate changes and their impacts on environmental risks even in case if the uncertainties are large and unknown. In the other words, it is better to prevent an accident than to deal with its consequences, which may be more expensive than the initial investment. In our study we present the new probabilistic hydrological model (and the method of its validation) allowing to perform a regional scale assessment of possible risks. These risks are accounted for by economic cost-lost models (see example by Räisänen and Palmer, 2001) even in case when the ways to evaluate the uncertainties are not yet found.

The text of the new version of the manuscript was corrected *p. 12, l. 24-30* (see above) and *p. 19, l. 16-17*: Räisänen, J. and Palmer T.N.: A probability and decision-model analysis of a multi model ensemble of climate change situations. Journal of Climate. 14: 3212–3226, 2001

Specific comments

L3-13: This new paragraph does not describe correctly the actual situation. The point is not if changes in climate drive changes in runoff, as this is obvious. The point is that nonstationary models require that we are sure about the future changes or, in other words, that the changes are deterministic (predictable). If the deterministic evolution (nonstationarity) of the process is only hypothetical, it is evident that we have no idea of how it will evolve, and supposed trends or regime shifts can simply be local fluctuations of perfectly stationary processes. In other words, we have hypothetical projections, but we have no idea of the actual evolution of climate and so runoff. Therefore, tools for nonstationarity are surely interesting, but their suitability in this context is highly questionable because they involve an additional source of uncertainty related to the unknown evolution pattern. This stresses once again the importance of assessing the uncertainty and reliability of design values.

Our response: we evaluate the uncertainties which are possible to evaluate in the frame of this study (see page  lines, page lines). Although the climate change is not deterministic and there is a lot of uncertainty in the future evolution of many variables in the climate system, climate change projections based on IPCC multi-model ensemble simulations show robust signals for the Eurasian Arctic. These include a large increase of air temperature and precipitation during this century (e.g., Collins et al., 2013; Laine et al., 2014). Due to the robustness of these model results, we think that the use of a nonstationary model is relevant, naturally bearing in mind the uncertainties listed on the first page of our response.

We have corrected the text on *p. 2, l. 20-30*: There are two opposite opinions about the climate changes and their effects to hydrological regime to answer the question: "Is it necessary to account

for climate changes by water managers and stakeholders?". According to Milly et al. (2008) the climate effects are already substantial, and should be taken into account by planers and water managers. The opposing view doubts a climate-driven changes and calls to attention the uncertainties due to short observed time series (Lins and Cohn, 2011; Montanari and Koutsoyiannis 2014; Serinaldi and Kilsby, 2015). We propose to account for the future climate changes and their impacts on environmental risks even in case if the uncertainties are unknown. In the other words, it is better to prevent an accident than to deal with its consequences, which may be more expensive than the initial investment. We consider that the changes in meteorological variables would remain noticed in runoff, which is an element of general water balance. From a practical point of view, a method to evaluate of extreme flood events based on climate scenarios is required, irrespective of debates about the extent predictability of the change.

"We consider it improbable that changes"… do you mean something like "We consider that it is improbable that changes"?? Please reword.
Our response: the sentence was reformulated in new text.

P3L4: please consider "physically-based model described by dynamic equations"
Our response: the word "dynamic" was excluded.

P3L4: please consider "the meteorological signal could be simulated by random generators." Do you mean "weather generators"? By the way, as I can see "a-priory" several time throughout the text, it is worth recalling that it should be "a priori": it is Latin and means "in (a) advance (priori)". If you are not familiar with such a kind of expressions please avoid them and use plain English.
Our response: yes, we mean the weather generators. The "a-priori" was replaced by "a priori" in the text twice. The text in the revised version of the manuscript was corrected on *p. 3, l. 4-7*: The stochastic components are incorporated into the physically-based hydrological model (Kuchment and Gelfan, 2011) to generate the flow time series based on the statistics of meteorological variables (weather generators). Thus, estimates of the extreme hydrological events (floods or droughts) with the required exceedance probability could be obtained for any climate scenario by producing the meteorological signal with Monte-Carlo method. Both approaches are usually applied for a single catchment.

P3L14-15: "the parameters of PDF are directly simulated from the meteorological mean values" please use correct terms: simulated means generate by e.g. a MC procedure. In this case, parameters are estimated as functions of covariates (meteorological mean values). Estimation and simulation have a different meaning and should be used in the correct context. "These parameters are further used to model PDF with theoretical distribution"??
Our response: the text was corrected to stress the difference between stochastic and probabilistic modeling on p. 3, l. 11-26: The approach presented in this paper could be named as probabilistic (to distinguish from the stochastic modelling described above). This approach allows us to skip the generation of the runoff time series, since only PDF parameters are directly calculated from the meteorological statistics for the projected periods of 20–30 years (Fig. 1). These simulated PDF parameters are further used to evaluate the future runoff values with the required exceedance probability using theoretical distributions i.e. from the Pearson system (Elderton et al., 1969). Since the probabilistic model simulates only three or four parameters of PDF, this approach allows to perform the regional-scale assessment of the detrimental hydrological events in the future, and to define the regions where the risks of damage to infrastructure is expected to increase.
In this study, we consider that for the Arctic, the maximal runoff is formed during a spring flooding. The Pearson type III distribution is used to model the PDF of the spring flood depth of runoff and to

estimate the maximal discharge with the required exceedance probability. The probabilistic approach used in this study combine the statistical methods and elements of the theory of Markov processes. Both of them are traditionally applied in hydrological engineering calculations to evaluate hydrological extremes (Kite, 1977; Benson, 1968; Kritsky and Menkel, 1946). The traditional analysis of flood and drought frequency requires hydrological time series to estimate the parameters of the PDFs. However, the parameters of PDFs can also be estimated from the statistics of meteorological variables.

P3L17: please replace "detrimental" with "extreme"
Our response: from our point of view, the term "detrimental" is more suitable in this sentence, as hydrological extremes are detrimental for human activities.

P3L34: "classical assumption used behind the engineering applications" please consider "classical assumption used in engineering applications"
Our response: the sentence is reformulated as suggested by the reviewer.

P4L34: "is calculated according method from"… do you mean "is calculated according to the method proposed (discussed) by (in)"
Our response: the sentence is reformulated as "is calculated according to the method proposed in...".

P4L36: Please consider "which reflects the water income to the catchment (due to melting) that affects the shape of hydrograph"
Our response: the sentence is reformulated as suggested by the reviewer.

P5L23: "based on"
Our response: corrected.

P5L28-30: what do you mean? Please reword in a more readable way
Our response: the paragraph was reformulated on *p. 5, l. 26-30*: The climate and runoff regimes are steady within both the reference and projected periods (the assumption of quasi-stationarity). The "steady" is defined statistically, i.e. there are no significant trends and changes in mean values of meteorological and hydrological characteristics within the periods. However, the basic statistics (mean, $C_v$ and coefficients of skewness $C_s$) are significantly different for the reference and projected periods.

P6L21-22: "different statistically significant PDF parameter values" do you mean that the difference between the parameter values in the two periods is statistically significant?
Our response: the sentence was reformulated to clarify this idea, p. 6, l.23-24: Two time periods should have the different PDFs' parameter values and this difference should be statistically significant (Kovalenko et al., 2010).

P6L25: "which allows performing the model ability to reproduce the measurements. In simplest case…" -> "which allows one to assess the model ability to reproduce the measurements. In the simplest case…"
and
P6L27-29: "Then, the training set is used to evaluate the model parameters, which are further used to calculate the modelling (or nominally predicted) dataset to compare with the testing/control set using chosen measure (the statistical goodness-of-fit tests in our case)." What does it mean "to calculate the modelling dataset"? Please reword using appropriate terms to describe corresponding

concepts.

Our response: the sentence was reformulated on p. 6, l. 26-30: In the simplest cross-validation procedure, the dataset of measurements (observations) is separated into two sub-sets, called the training set and the testing/control set. The training set is used to evaluate the model parameters, which are further used to calculate the nominally predicted values of the parameters of the control PDFs. In our case, the nominally predicted PDF was compared with the empirical distribution for the testing/control set using statistical goodness-of-fit tests.

P9L20: Chylek et al. (2011) was not accepted for final publication. Please provide a published reference.

Our response: Chylek et al. (2011) was replaced by von Salzen, K., J. F. Scinocca, N. A. McFarlane, J. Li, J. N. S. Cole, D. Plummer, D. Verseghy, M. C. Reader, X. Ma, M. Lazare, L. Solheim: The Canadian Fourth Generation Atmospheric Global Climate Model (CanAM4). Part I: Representation of Physical Processes, Atmosphere-Ocean, 51, 104-125, doi:10.1080/07055900.2012.755610, 2013.

P9L22: "the the"
Our response: corrected.

P10L13: "do no" replaced by "do not"
Our response: corrected.

P10L26: do you mean "especially"? Please reword the sentence.
Our response: corrected.

P10L26: "to calculate the maximal runoff from tailed values with required probability ofexceedance" -> "to calculate the yearly maximum runoff with required probability of exceedance". Please use homogenous terms to describe the same object, avoiding e.g. "maximal runoff" for what is previously defined as "yearly maximum runoff" and brand new terms like "tailed values" or "maximal extremes" to denote "extreme quantiles" or simply "extreme values".
Our response: the terms consistency is checked.

P13L18: "toevaluate"
Our response: corrected.

We thank to the Reviewer for the discussion and comments, which were useful to present the result of our study clearly and consistently.

**Response to the Review on**
**"Assessment of extreme flood events in changing climate for a long-term planning of socio-economic infrastructure in the Russian Arctic"**
**by E. Shevnina et al.**

We thank Dr. Irina Fedorova for the useful comments, and further we provide our answers and responses (the blue text) to her questions and comments (the black text).

**General comments**
The last version of the paper "Assessment of extreme flood events in changing climate for a long-term planning of socio-economic infrastructure in the Russian Arctic" by E. Shevnina et al. has been forcefully improved. The new supposed method of a probabilistic model using for the runoff prediction are clarified for readers. The figure 1, where three approaches of climate change evaluation on hydrological processes, is also helpful for building of the new model place in the list of well-known types of models. Received results will be fruitful for stakeholders in this interpretation that explained in the article. The regional map (figure 7) with predicted substantial changes of a spring flood depth of runoff value is an important result for engineering tasks in a permafrost zone of the Arctic. Figure 8 gives possibility to look better on distribution of measured points in the Arctic.
Nevertheless there are some comments for the method that are offered to be discussed in the last chapter.

Supposed in the paper probabilistic model consists a precipitation amount per year as input information. It is a consolidate income without seasonal dynamic. Such kind of approach can be dramatically insufficient. Seasonal distribution of volume of precipitation is more important than average annual one: if the most part of precipitation fall in one day it is absolutely differ than the same volume of precipitation during three months, for example. There is a special flood coincidence factor $k_0$ in the equation (1). But authors do not use it ($k_0$=1 – Page 12, Line24 as an example*)* - they summarise all precipitation (rain, snow, groundwater) and put time τ=1 (Page 14, Line 30-31). From one side, it can give possibility to receive an overestimated result of runoff or lead receiving an average result, from another side. Supposed coefficient $\bar{c}$ (Page 5, Line 20) will not help to solve the problem of catchment heterogeneousness.
How the authors see this problem? How they will solve it? There is no doubt that it should be solved exactly in this article. Some time and work are needed for this. The chapter could be considerably supplemented with discussion about it. Otherwise, the chapters 3 and 4 look like a list of received results.
Our response: from our point of view, two questions were raised by the reviewer within this comment: (i) how to evaluate precipitation amount to provide the forcing for the probabilistic model of the spring flood runoff and (ii) how to set the special flood coincidence factor $k_0$ in the Eq. 1 for the projected period.
(i). It is obviously that spring floods are formed by a snow melting plus a rainfalls during flooding period. These values can be estimated from observations: only daily values of air temperature, precipitation and river discharge are required to define the duration of cold and spring flooding periods. However, it rises challenge in setting of the forcing climatology for the projected periods since the climate scenarios do not provide statistics connected with duration of spring flooding in future. Thus, the mean values of precipitation amount can be practically estimated over whole year or only for cold period. The simple correlation analysis shows that the relationships between spring flood depth of runoff and precipitation evaluated over whole year and cold period only are similar for the northern territories (Shevnina, 2011).

*p. 9, l.13–15.* For the precipitation we use the annual values although the spring floods are formed only by a snowfall and spring rainfall. However, in the Arctic the relationships between spring flood depth of runoff and annual and winter-spring sums of precipitation are similarly strong (Shevnina, 2011).

*p. 19, l. 30–32.* Shevnina, E.V.: The relationships between an annual and winter precipitation amount and flooding runoff on the rivers over the Russian Arctic. Scientific reports of the Russian State Hydrometeorological University, 20, 6–12, 2011. (in Russian).

(ii). Originally, the equation (1) suggested to apply for ungauged catchments to evaluate maximal discharges from flood flow depth (for values with particular exceedance probability). The value of spring flood coincidence factor of $k_0$ is regional, and it is estimated from observations on neighboring gauge (SP33-101-2003, 2004). The $k_0$ depicts a simultaneousness of water input, i.e. how this input affects to river maximal discharges, and $k_0$ does not reflect fraction of different precipitation types (rain, snow, groundwater). In our study, the value of $k_0$ was set as constant for the reference and projected periods. The $k_0$ for the future periods can evaluated by physically-based model for the region with substantial changes allowing generation of steam-flow hydrographs. The value of $k_0$ is equal to 1 in the example of the Nadym River at Nadym City, and it was done for sake of simplicity.

Several regions in the Russian Arctic noticed in the text (Figure 5, for example), list of considered rivers presented in the table 1 and 2. But the region specific had not been hydrological / hydrographically explained (how the authors sort them out?).

Our response: The Russian Arctic was divided into four parts to develop the specific parameterization schemes since the uniform scheme provides less effective results. The principles of the regions division was based on the dominant geology, geomorphology and permafrost features and were presented in Shevnina (2012).

The regions consider different quantity of measured points (Figure 8)? There are 3 points for Kola region, 10 – for Arkhangelsk and Komi, 3 – for West Siberia, and 6 – for a huge East Siberia. Does the East Siberia have enough density or not in your opinion?

Our response: the points in Fig. 8 present the gauges used for the model validation. These gauges were selected from other (76 gauges) since two periods with statistically significant difference in the mean values in the runoff timeseries were defined (see Section 2.2). The amount of gauges is limited especially in Siberia, but these data were not used to calculate the projected runoff's statistics.

How do you expand all parameters and results on whole territories without measured points?

Our response: the model parameters were estimated in the grid nodes of particular climate model (p. 9, lines 29-31) using the Eq. 3 based on the reference climatology/hydrology (Fig. 4). In this procedure the climatology were extracted from the interpolated fields, and the maps provided by Vodogretskiy (1986) and Rogdestvenskiy (1986). We did not extrapolate the model parameters obtained for the particular watersheds on the stage of the model validation.

How the extrapolation has been done for the map building? Discussion about validity of extrapolation for the West and East Siberia regions could be added in the chapter 3 (or to the methods chapter).

Our response: the extrapolation of the mean values and CV of spring flood flow depths were done by Vodogretskiy and Rogdestvenskiy (1986), and we did not any extrapolation of runoff statistics. These maps are included in the Annex for the Guideline for engineers to evaluate maximal discharges of exceedance probability for building projects from 1986 to 2004 (the actual Guideline SP33–101–2003 was published, 2004).

The obtaining data procedure is already described on the page 9, lines 5–14, page 9, lines 29–37 and page 10, lines 1–2.

In the revised version of the manuscript the following corrections were done:
*p.4 line 34–35:* where $k_0$ is flood coincidence factor, which reflects a simultaneousness of precipitation/melting water input, i.e. depend on shape of the hydrograph,
*p.5, line 3-5:* For ungauged basin the value of $k_0$ is estimated from observations on a neighboring gauge located on a same type of landscape (SP33-101-2003, 2004). In our study, the value of $k_0$ was considered to be constant for the reference and projected periods.
*p.12, line 32-34:* the value of $k_0$ was considered to be constant for the reference and projected periods and it was set to be equal to 1 in our example for sake of simplicity.
*p.14, line 20-21:* Such models also allow to evaluate the values of spring flood coincidence factor $k_0$ (Eq. 1) for the projected periods (which was constant in our calculations).
*p. 9, line 14-19:* The mean values and $C_v$ of the spring flood depth of runoff were extracted to the model grid nodes from the maps (Rogdestvenskiy, 1986; Vodogretskiy, 1986). In our study no observations of multi-year runoff were used to evaluate the mean value and $C_v$ for the reference period and no extrapolation was applied for the regions without observations.
*p. 13 line 24-25*: (iv) to estimate the mean values, $C_v$ and $C_s$ from the observed time series of runoff or to evaluate them from the regional maps (i.e. Spence and Burke, 2008)…
*p. 19, l. 37-38*: Spence, C. and A. Burke, 2008, "Estimates of Canadian Arctic Archipelago Runoff from Observed Hydrometric Data," Journal of Hydrology, 362, 247-259.

**Specific comments**
Page 1, Line 25-28. Authors enumerate socio-economic projects for different regions. If the titles or/and web-pages of these projects will be included (or reference for them) it could be helpful for readers and stakeholders.
Our response: the following links were added into the text of new version of the manuscript:
*page 1, line 26-28*: the Mackenzie Valley, Canada: *http://www.mackenziegasproject.com*; the Prudhoe Bay, USA: *http://petrowiki.org/Prudhoe_Bay_field*; for the Pechora and Yamal, Russia: *http://www.gazprom.com/about/production/projects/mega-yamal/*.

Page 1, Line 28-30. Only flood-related risks are considered in the paper. Even though ice jams and sub riverbed taliks are not included in the authors' method like a parameter. It should be discussed or marked in the discussion chapter for clear understanding the situation with economical necessities for such projects as bridges, for example.
Our response: in the revised manuscript we stress that the model used in this study does not account for the flood extremes originating due to ice jams and tides/surges (*page 10, line 5-6*).

Page 2, Line 14-17. Authors mark the increase for precipitation (according to some references) and then add the explanation about "…high greenhouse gas concentration pathway…". Why only this reason marked? What about air temperature cycling? There were a lot of cycles with high or low temperature –without greenhouse gas effect on the Earth climate. That is quite questionable thing. Authors have a reference on the IPCC report, but this reason is suggested to be deleted from the text any way.
Our response: there probably is misunderstanding of the Reviewer about "greenhouse gas concentration pathway": we did not discuss the reason of the precipitation increasing. On page 2, line 16, the "greenhouse gas concentration pathways" are the name of the climate scenarios according Fifth IPCC Report.

Page 5, Line 20, Line 22. The explanation of $k$ – the runoff coefficient and $G_N$ should be added.

Our response: the following explanations were added to *p. 5, l. 21–22*: *k,* (which is a dimensionless coefficient, the ratio of the amount of runoff to the amount of precipitation received) and *p. 5, l. 24:* $G_N$ is the variance of the annual precipitation amount.

Page 6, Line 9. The short explanation of differences between approaches of Kovalenko and Shevnina should be added in the text.
Our response: the text of the new version of the manuscript was improved:
*p. 6, l. 10-12*: The values of the parameters $\bar{c}$ and $G_{\tilde{N}}$ either can be set constant for the projected time period as proposed by Kovalenko et al. (2010) or depending on the future climatology. To evaluate the projected values of the parameter $\bar{c}$ depending on the average precipitation and air temperature, the linear equations are suggested by Shevnina (2012).

Page 8, Line 24. The "personal communication…" with Bryazgin is suggested to be moved to the acknowledgments. There is no reference for him in the list.
Our response: we would like to refrain to remove the reference to the personal communication since it is common practice to use it as a reference.

Page 12, Line 8. The Nadym River is located not in the "…southern part of Western Siberia…" Nadym is the Arctic River. May be authors mean the Western Siberia considering region of the picture 4 and 7?
Our response: the text was corrected as follows:
*p. 12, line 11-12*: The Nadym River is located in the region, where an increase of mean values of the spring flood flow depth was predicted under RCP2.6 scenario (Fig. 7, right, upper plot).

Page 12, Line 18. Even though the second statistical moment is $C_v$ unification in the text should be checked better. Change $C_v$ in the bracket to $m_2$, please.
Our response: it was corrected in the revised text.

Page 12, Line 27-28. The reference for the Nadym River discharge 573 m$^3$ s$^{-1}$ should be added.
Our response: the value of 573 m$^3$ s$^{-1}$ is the difference between the historical and projected discharges. This detail now is provided in the text of the revised manuscript in *p. 13, l. 2*.

Page 13, Line 7-8. The examples of datasets should be removed from this chapter to 2.3, for example, because these are not the authors' results.
Our response: The examples of datasets (as well as *p. 13, l. 17-21*) were added due to suggestion of the other Reviewer. We would refrain to remove both paragraphs.

Page 15, equation (A5). There are three statistical moments in the equation. But in the chapter 2.1 of the text marked only two of them. It should be brought to conformity.
Our response: the additional explanation is now presented after equation (A5) on page 16, and the last equation in the Annex is the same as the Eq. (2) on page 5.

**Technical corrections**
Page 1, Line 23. In the end of sentence – "sho**u**ld be…"
Page 4, Line 37. The explanation of parameter $\mu$ should be removed before $h_p$ - to keep the order as in the equation (1) is.
Page 5, Line 33; Page 6, Line 4 – to add abbreviation (*r*) to the "…chosen reference (*r*) period…" – for better understanding; and (*f*) – for "…for future (*f*) values…"
Page 9, Line 22. Double "the" – cut one of it, please.
Page 13, Line 18. The divide "…toevaluate…", please.
Our response: the technical corrections were done in the revised text according to the list above.

Table 2. The index "*f*" should be added to the Notations for corresponding parameters.
Our response: the text was corrected.

Figure 2. The explanation for points "A", "S","E" and "i" should be added.
Our response: the comments to the figure were corrected.

Figure 4. Dimensions for each scale should be added because they are different.
Our response: the figure was corrected.

We hope that the changes made help to better understand the results of our study.

[revised manuscript text omitted]

---

## Author Response (AR3)

Dear Editor,

Thank you for the good news about our manuscript, which was revised again to fix the rest of typing mistakes and to fit the text to the rules of the HESS.

The minor text correction was applied for the manuscript structure (the method and data section) and sentences (to exclude the typing mistakes), two more references were also added to the list of references (Madsen et al., 2013 and Pugachev et al., 1974) and the figures (2, 3, 4, 5, 6, 8) were corrected.

with the best regards
Authors

---

## Author Response (AR4)

Dear Editor,

The manuscript was now revised by the native speaker, and the text was improved. Also, the list of references was corrected according to the rules of HESS.

With the best regards,

Authors